# Hitchhiker's guide on the relation of Energy-Based Models with other generative models, sampling and statistical physics: a comprehensive review

**Davide Carbone**                                                      *davide.carbone@phys.ens.fr*
*Laboratoire de Physique de l'Ecole Normale Supérieure, ENS Université PSL,*
*CNRS, Sorbonne Université, Université de Paris, Paris*

**Reviewed on OpenReview:** *https://openreview.net/forum?id=VTgixSbrJI*

## Abstract

Energy-Based Models have emerged as a powerful framework in the realm of generative modeling, offering a unique perspective that aligns closely with principles of statistical mechanics. This review aims to provide physicists with a comprehensive understanding of EBMs, delineating their connection to other generative models such as Generative Adversarial Networks, Variational Autoencoders, and Normalizing Flows. We explore the sampling techniques crucial for EBMs, including Markov Chain Monte Carlo (MCMC) methods, and draw parallels between EBM concepts and statistical mechanics, highlighting the significance of energy functions and partition functions. Furthermore, we delve into recent training methodologies for EBMs, covering recent advancements and their implications for enhanced model performance and efficiency. This review is designed to clarify the often complex interconnections between these models, which can be challenging due to the diverse communities working on the topic.

## 1 Introduction

Several surveys on Energy-Based Models (EBMs) have been published in recent years, each offering valuable insights into specific aspects of this modeling framework. Some works focus on their historical development and original formulation prior to the deep learning era (LeCun et al., 2007), while others examine their conceptual ties with statistical physics and the theory of Boltzmann Machines (Huembeli et al., 2022), or their evolving role in the modern generative modeling landscape (Bond-Taylor et al., 2021; Song & Kingma, 2021). However, these reviews often lean towards either algorithmic and empirical advances from a machine learning perspective or foundational insights rooted in statistical mechanics, rarely addressing both comprehensively. Moreover, crucial notions such as sampling, energy normalization, and the interpretation of the Boltzmann-Gibbs distribution are frequently treated as technical details or relegated to side discussions, despite being central to both the theory and practical implementation of EBMs.

In contrast, the present review aims to provide a unified and pedagogical account of Energy-Based Models that is both self-contained and accessible to a broad interdisciplinary audience. Specifically, we take the viewpoint that EBMs should not be understood solely as a machine learning model class defined by an unnormalized density function, but rather as a natural interface between statistical physics, sampling theory, and modern generative modeling. Our goal is to elevate the role of sampling—from a peripheral computational tool to a central component of the model's theoretical foundation and training dynamics—emphasizing how concepts from nonequilibrium statistical physics, such as Langevin and Metropolis-Hastings dynamics, directly shape the behavior and feasibility of EBMs.

This work is not just a tutorial on existing training algorithms; rather, it seeks to clarify the fundamental principles and limitations that stem from the very definition of EBMs as energy-parametrized distributions. In doing so, we aim to help readers understand why tasks such as likelihood evaluation, generation, or

inference remain non-trivial within this framework. This perspective also allows us to critically assess the relationships between EBMs and other generative models—including normalizing flows, GANs, VAEs, and diffusion models—highlighting both structural differences and shared algorithmic tools (e.g., score-based learning, Langevin sampling).

We believe that such an integrative approach is currently lacking in the literature. While many excellent surveys focus on specific algorithmic developments or empirical applications, few attempt to articulate an auto-consistent conceptual framework that situates EBMs at the intersection of physics, sampling, and generative modeling. This review aspires to fill that gap, acting as a "Hitchhiker's Guide" for physicists venturing into generative modeling, and for machine learning researchers interested in the deep physical analogies underlying their models, trying to balance soundness for newcomers and technical details.

To provide historical depth and context, we also include in Appendix A a brief chronology of the development of EBMs and their connections to key ideas in both physics and AI. While not necessary for understanding the main technical content, we strongly encourage readers to consult it for a broader perspective.

**Contribution.** The structure of the review will be the following:

- in Section 2 we define Energy-Based Models and we highlight the main difficulty related to its training through cross-entropy minimization;

- in Section 3 we present a review of the principal generative models as opposed to EBMs. Then, we conclude the section with a comparative scheme between all the presented models, with an accent on the unique features of EBMs;

- in Section 4 we review the main MCMC methods used to sample from a Boltzmann-Gibbs ensemble, hence used in the context of EBM training to generate the necessary sample used to perform gradient descent on cross-entropy;

- in Section 5 we summarize the derivation of Boltzmann-Gibbs equilibrium ensemble, motivating the importance of concepts as free energy in the context of statistical learning;

- in Section 6 we present recent progress in EBM training and we highlight their limitation, in relation with relatively old methods as Constrastive Divergence.

## 2   Definition of EBM

In this Section, we provide the basic formal definition of Energy-Based Model. We will adopt the notation and the presented assumption throughout the present work. First of all, the problem we consider can be formulated as follows: we assume that we are given $n \in \mathbb{N}$ data points $\{x_i^*\}_{i=1}^n$ in $\mathbb{R}^d$ drawn from an unknown probability distribution that is absolutely continuous with respect to the Lebesgue measure on $\mathbb{R}^d$, with a positive probability density function (PDF) $\rho_*(x) > 0$ (also unknown). This is a standard problem in statistical learning, where *learning from data* here refers to the ability to fit the data distribution and to generate new examples. More precisely, our aim is to estimate $\rho_*(x)$ via an energy-based model (EBM), i.e. to find a suitable energy function in a parametric class, $U_\theta : \mathbb{R}^d \to [0, \infty)$ with parameters $\theta \in \Theta$, such that the associated Boltzmann-Gibbs PDF

$$\rho_\theta(x) = Z_\theta^{-1} e^{-U_\theta(x)}; \qquad Z_\theta = \int_{\mathbb{R}^d} e^{-U_\theta(x)} dx \qquad (1)$$

is an approximation of the target density $\rho_*(x)$. Actually, any probability density function can be written as a Boltzmann Gibbs ensemble for a particular choice of $U(x)$. The normalization factor $Z_\theta$ is known as the partition function in statistical physics (Lifshitz & Pitaevskii, 2013), see also Section 5, and as the evidence in Bayesian statistics(Feroz & Hobson, 2008).

**Remark 2.1.** *Even if $U_\theta$ is known, an explicit analytical computation of the partition function is generally unfeasable. If the dimension $d$ is big enough, the integral defining $Z_\theta$ cannot be computed using standard quadrature methods. The only possibility is Monte-Carlo sampling(Liu, 2001). To employ such method, one*

can express the partition function as an expectation $\mathbb{E}_0$ with respect to a chosen probability density function $\rho_0$, i.e.

$$Z_\theta = \mathbb{E}_0 \left[ \frac{e^{-U_\theta}}{\rho_0} \right] \tag{2}$$

*The selected density must be known pointwise in $\mathbb{R}^d$, including the normalization constant, and it should be easy to sample from. If these conditions are met, one can compute the partition function by simply replacing the expectation in equation 2 with the corresponding empirical average computed using samples drawn from $\rho_0$. Unfortunately, finding a probability density that satisfies these properties is challenging. For a general choice that is not tailored to $e^{-U_\theta}$, the estimator is likely to be very poor, characterized by a very large, or even infinite, coefficient of variation.*

One advantage of EBMs is that they provide generative models that do not require the explicit knowledge of $Z_\theta$. In Section 4 we will review some routines that can in principle be used to sample $\rho_\theta$ knowing only $U_\theta$ – the design of such methods is an integral part of the problem of building an EBM.

To proceed we need some assumptions on the parametric class of energy:

**Assumption 2.1.** *For all $\theta \in \Theta$:*

1. *$U_\theta \in C^2(\mathbb{R}^d)$;     $\exists L \in \mathbb{R}_+ : \|\nabla\nabla U_\theta(x)\| \leq L \quad \forall x \in \mathbb{R}^d$;*

2. *$\exists a \in \mathbb{R}_+$ and a compact set $\mathcal{C} \in \mathbb{R}^d : \; x \cdot \nabla U_\theta(x) \geq a|x|^2 \quad \forall x \in \mathbb{R}^d \setminus \mathcal{C}$.*

The need for the first assumption will be discussed in Section 4: it is related to wellposedness and convergence properties of the dynamics used for sampling, i.e. Langevin dynamics and its specifications. The second assumption guarantees that $Z_\theta < \infty$ (i.e. we can associate a PDF $\rho_\theta$ to $U_\theta$ via equation 1 for any $\theta \in \Theta$). We provide now two important definitions:

**Definition 2.1** (Convexity). *A function $\varphi : \mathbb{R}^d \to (-\infty, +\infty]$ is convex if given $0 < \lambda < 1$ and $x_1, x_2 \in \mathbb{R}^d$ such that $x_1 \neq x_2$, the following is true*

$$\varphi(tx_1 + (1-t)x_2) \leq t\varphi(x_1) + (1-t)\varphi(x_2) \tag{3}$$

**Definition 2.2** (Log-Concavity). *A density function $\rho$ with respect to Lebesgue measure on $(\mathbb{R}^d, \mathcal{B}^d)$ is log-concave if $\rho = e^{-\varphi}$ where $\varphi$ is convex.*

A non-convex function could have more than one local but not global minima; conversely, a non-log-concave probability density could have more than local maxima, which are called *modes*. It is important to stress that Assumption equation 2.1 *does not* imply that $U_\theta$ is convex (i.e. that $\rho_\theta$ is log-concave): in fact, we will be most interested in situations where $U_\theta$ has multiple local minima so that $\rho_\theta$ is multimodal. We will elaborate on the topic in Section 4. It is well known as for optimization problems, non-convex cases are the most complicated. Similarly, sampling from a non-log-concave probability density function (PDF) can be extremely challenging. Another assumption we will adopt is:

**Assumption 2.2.** *Without loss of generality $\exists\theta_* \in \Theta \; : \; \rho_{\theta_*} = \rho_*$, that is $\rho_*$ is in the parametric class of $\rho_\theta$.*

The aims of EBMs are primarily to identify $\theta_*$ and to sample $\rho_{\theta_*}$; in the process, we will also show how to estimate $Z_{\theta_*}$.

**Example 2.1.** *Let us present a simple example to visualize the relation between convexity and log-concavity. In Figure 1 we plot side by side the PDF of a Gaussian mixture in 1D and the associated potential*

$$U_\theta(x) = \log\left[ p\exp\left(-\frac{(x-\mu_1)^2}{\sigma_1^2}\right) + (1-p)\exp\left(-\frac{(x-\mu_2)^2}{\sigma_2^2}\right) \right] \tag{4}$$

*where $\theta = \{p, \mu_{1,2}, \sigma_{1,2}\}$. The specific values are $p = 0.7$, $\mu_1 = 0$, $\mu_2 = 5$, $\sigma_1 = 1$ and $\sigma_2 = 0.5$. It is clear the correspondence between minima of $U_\theta(x)$ and maxima, that is modes, of $\rho$.*

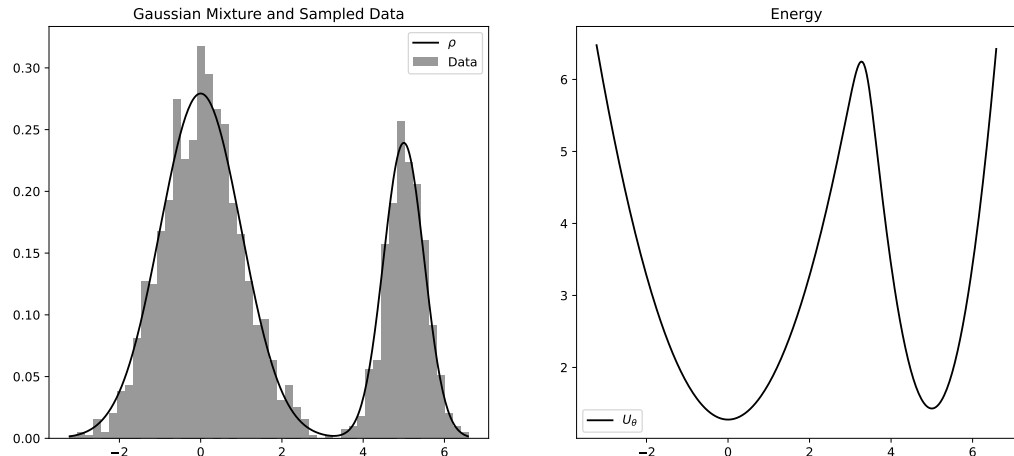

Figure 1: *Gaussian Mixture.* Plot of PDF with sampled histogram and associated energy $U_\theta$.

## 2.1 Cross-entropy minimization

Once we defined an EBM, we need to measure its quality with respect to the data distribution. Possibly, this would provide a way to train its parameters. Hence, we define some important quantities:

**Definition 2.3.** *Consider two probability densities on $\mathbb{R}^d$ and absolutely continuous with respect to Lebesgue measure, namely $\rho_1$ and $\rho_2$. We define*

1. **Cross Entropy**

$$H(\rho_1, \rho_2) = - \int_{\mathbb{R}^d} \log \rho_2(x) \rho_1(x) dx \tag{5}$$

2. **Kullback-Leibler divergence(Kullback & Leibler, 1951)**

$$D_{KL}(\rho_1 \parallel \rho_2) = \int_{\mathbb{R}^d} \rho_1(x) \ \log\left(\frac{\rho_1(x)}{\rho_2(x)}\right) \ dx \tag{6}$$

3. **Entropy**

$$H(\rho_1) = - \int_{\mathbb{R}^d} \log \rho_1(x) \rho_1(x) dx \tag{7}$$

The KL divergence is a widely used estimator for the dissimilarity between probability measures. It satisfies the non-negativity condition

$$D_{\mathrm{KL}}(\rho_1 \parallel \rho_2) \geq 0, \qquad D_{\mathrm{KL}}(\rho_1 \parallel \rho_2) = 0 \iff \rho_1 = \rho_2 \text{ a.e.} \tag{8}$$

However, it is not a proper distance since it is not symmetric and it does not satisfy triangular inequality. The following trivial lemma relates the three quantities we introduced in Definition 2.3:

**Lemma 2.1.** *The following equality holds for any choice of PDFs $\rho_1$ and $\rho_2$*

$$H(\rho_1, \rho_2) = H(\rho_2) + D_{KL}(\rho_2 \parallel \rho_1) \tag{9}$$

One can also use the cross-entropy of the model density $\rho_\theta$ relative to the target density $\rho_*$ as an estimate of diversity between the two PDFs; in such case, 5 simplifies becoming

$$H(\rho_*, \rho_\theta) = \log Z_\theta + \int_{\mathbb{R}^d} U_\theta(x) \rho_*(x) dx \tag{10}$$

Because of 9, the difference between the cross-entropy and the KL divergence is $H(\rho_*)$, a term that depends just on the data distribution. Hence, the optimal parameters $\theta^*$ are solution of an optimization problem on $\Theta$, namely

$$\theta^* = \underset{\theta \in \Theta}{\arg\min}\, D_{\mathrm{KL}}(\rho_* \| \rho_\theta) = \underset{\theta \in \Theta}{\arg\min}\, H(\rho_*, \rho_\theta), \tag{11}$$

meaning that the entropy of $\rho_*$ plays no active role in solving such minimization problem. There is a subtle issue in this reasoning: unlike KL divergence, the cross-entropy is not bounded from below, and in particular $H(\rho, \rho) := H(\rho) \neq 0$. That is, we should compute $H(\rho_*)$ to estimate the minimum value of cross-entropy. Unfortunately, most of the empirical estimators to be used when $\rho_*$ is known through samples suffer in high dimension(Ao & Li, 2023). Solving equation 11 is equivalent to maximum likelihood method, a widely used practice in parametric statistics(Stigler, 2007).

The use of cross-entropy avoids the very problematic computation of $H(\rho_*)$, but in 10 the estimation of $Z_\theta$ is also needed. However, the most common routines for cross-entropy minimization are gradient-based: they rely on the gradient of $\partial_\theta H(\rho_*, \rho_\theta)$ and not on the cross-entropy itself. The former can be computed using the identity $\partial_\theta \log Z_\theta = - \int_{\mathbb{R}^d} \partial_\theta U_\theta(x) \rho_\theta(x) dx$, obtaining

$$\begin{aligned} \partial_\theta H(\rho_*, \rho_\theta) &= \int_{\mathbb{R}^d} \partial_\theta U_\theta(x) \rho_*(x) dx - \int_{\mathbb{R}^d} \partial_\theta U_\theta(x) \rho_\theta(x) dx \\ &:= \mathbb{E}_*[\partial_\theta U_\theta] - \mathbb{E}_\theta[\partial_\theta U_\theta]. \end{aligned} \tag{12}$$

This is a crucial expression for the present work, and the consequence is immediate:

**Remark 2.2** (Fundamental problem for EBM training). *Estimating $\partial_\theta H(\rho_*, \rho_\theta)$ requires calculating the expectation $\mathbb{E}_\theta[\partial_\theta U_\theta]$. In contrast $\mathbb{E}_*[\partial_\theta U_\theta]$ can be readily estimated on the data.*

Typical training methods, e.g. based on the so-called Constrastive Divergence(Hinton, 2002) and its specifications (see Subsection 6), resort to various approximations to calculate the expectation $\mathbb{E}_\theta[\partial_\theta U_\theta]$. While these approaches have proven successful in many situations, they are prone to training instabilities that limit their applicability. The cross-entropy is more stringent, and therefore better, than objectives like the Fisher divergence used to train other generative models: for example, unlike the latter, it is sensitive to the relative probability weights of modes on $\rho_*$ separated by low-density regions (Song & Kingma, 2021).

## 3 EBMs among generative models

In this section, we briefly outline the main generative models, aiming to provide a general framework without focusing on technical details. As introduced earlier, a generative model is a computational tool that produces new instances representative of a dataset—for example, generating new images of dogs after training on a dataset of dog images. Evaluating the quality of generated data is often non-trivial, and designing a good generative model typically involves a trade-off between ease of training and ease of generation.

Historically, generating data meant collecting measurements. The advent of computer simulations, such as the pioneering Fermi-Pasta-Ulam-Tsingou experiment (Fermi et al., 1955), marked a shift toward generating synthetic data from models. In statistics, this process is referred to as "sampling" (see Section 4). Today's generative AI, enabled by vast computational resources and data availability, continues this shift by generating data from data. As with the early internet[1], what began in research now shapes everyday life. Generative Pre-Trained Transformers like ChatGPT exemplify this trend. These tools are prompting broader discussions about artificial general intelligence and its societal implications (Morozov, 2023; Fjelland, 2020; Federspiel et al., 2023).

We now turn to the core families of generative models, and ultimately, we will highlight their connections to Energy-Based Models.

### 3.1 Variational Autoencoders

As the name suggests, to introduce variational autoencoders (VAEs) (Girin et al., 2020), we first recall the structure of a standard autoencoder (AE) (Hinton & Salakhutdinov, 2006), illustrated in Figure 2. An

---

[1]https://www.livinginternet.com/i/ii_arpanet.htm

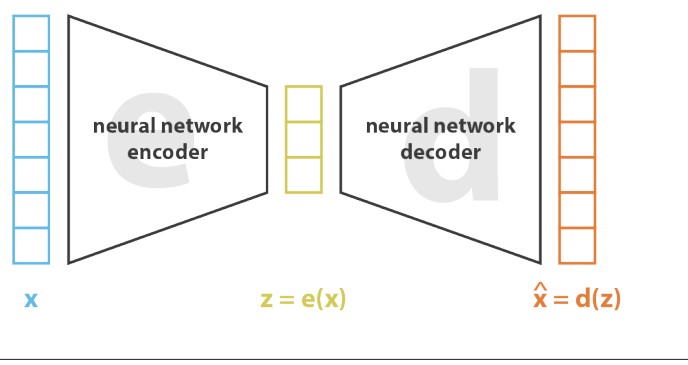

$$\text{loss} \; = \; || \, x - \hat{x} \, ||^2 \; = \; || \, x - d(z) \, ||^2 \; = \; || \, x - d(e(x)) \, ||^2$$

Figure 2: Representation of an autoencoder, taken from `https://towardsdatascience.com/understanding-variational-autoencoders-vaes-f70510919f73`

AE is a deep neural network composed of two parts: an encoder $e(x)$, which maps the input $x \in \mathbb{R}^d$ to a latent representation $z \in \mathbb{R}^L$ with $L \leq d$, and a decoder $d(z)$, which attempts to reconstruct $x$ from $z$. Training minimizes the reconstruction error between $x$ and $d(e(x))$. Once trained, AEs can be used as compression tools, and additional layers can be applied in latent space for downstream tasks (Bank et al., 2023). This process is entirely deterministic. Variational Autoencoders (Kingma & Welling, 2014) extend AEs by introducing probabilistic encoders and decoders, respectively referred to as inference and generative models. Let $\rho_\theta(x,z) = \rho_\theta(x|z)\rho(z)$ be a joint distribution on $\mathbb{R}^d \times \mathbb{R}^L$, with prior $\rho(z) = \mathcal{N}(z; \mathbf{0}_L, \boldsymbol{I}_L)$ and decoder defined by a conditional Gaussian:

$$\rho_\theta(x|z) = \mathcal{N}(x; \boldsymbol{\mu}_\theta(z), \mathrm{diag}\{\boldsymbol{\sigma}_\theta^2(z)\}), \tag{13}$$

where the mean and variance are neural network outputs. Alternatives exist for other data types, e.g., audio (Girin et al., 2019).

The marginal model distribution is given by

$$\rho_\theta(x) = \int_{\mathbb{R}^L} \rho_\theta(x|z)\rho(z)\, dz, \tag{14}$$

and training aims to minimize the KL divergence from the true data distribution $\rho_*(x)$:

$$\theta^* = \arg\max_{\theta \in \Theta} \mathbb{E}_*[\log \rho_\theta(x)], \tag{15}$$

analogously to EBMs. However, the marginal likelihood equation 14 is generally intractable, preventing direct evaluation. To address this, VAEs rely on a variational approximation, which we introduce next:

**Definition 3.1** (ELBO). *Let $\mathcal{F}$ denote a variational family defined as a set of PDFs over the latent variables $z$. For any $q(z) \in \mathcal{F}$, the* **Evidence Lower Bound (ELBO)** *(also known as* variational free energy*) $\mathcal{L} : \Theta \times \mathcal{F} \times R^d \to \mathbb{R}$ is defined as*

$$\mathcal{L}(\theta, q(z); x) = \mathbb{E}_{q(z)}[\log \rho_\theta(x,z) - \log q(z)] \tag{16}$$

**Lemma 3.1.** *The following properties hold true:*

*1. Decomposition of marginal log-likelihood(Neal & Hinton, 1998).*

$$\log \rho_\theta = \mathcal{L}(\theta, q(z); x) + D_{KL}(q(z) \parallel \rho_\theta(z|x)) \tag{17}$$

*2. Bound on marginal log-likelihood.*

$$\mathcal{L}(\theta, q(z); x) \leq \log \rho_\theta(x)$$
$$\mathcal{L}(\theta, q(z); x) = \log \rho_\theta(x) \iff q(z) = \rho_\theta(z|x) \tag{18}$$

*Proof.* The proof of (1) is trivial:

$$\mathcal{L}(\theta, q(z); x) + D_{\mathrm{KL}}(q(z) \| \rho_\theta(z|x)) = \mathbb{E}_{q(z)}[\log \rho_\theta(x, z) - \log q(z)]$$
$$+ \mathbb{E}_{q(z)}[\log q(z) - \log \rho_\theta(z|x)] = \mathbb{E}_{q(z)}\left[\log\left(\frac{\rho_\theta(x, z)}{\rho_\theta(z|x)}\right)\right] = \log \rho_\theta(x) \tag{19}$$

where we used the definition of conditional probability and the fact that the expectation is computed in the latent space. (2) is a direct consequence of equation 17 since the KL divergence is non-negative and identically zero just when $q(z) = \rho_\theta(z|x)$. □

Thanks to these results, the log-likelihood can be estimated via the Expectation-Maximization (EM) algorithm (Dempster et al., 1977): the (E) step solves the unconstrained variational problem at fixed $\theta$

$$q_*(z) = \arg\max_{q \in \mathcal{F}} \mathcal{L}(\theta, q(z); x), \tag{20}$$

while the (M) step maximizes the ELBO w.r.t. $\theta$ at fixed $q(z)$. Since $q(z)$ depends on $x$, it is more precisely written as $q(z|x)$. Under suitable conditions, this EM procedure converges to a maximum and saturates the bound in equation 18.

However, solving this functional optimization can be intractable. A practical alternative is *fixed-form variational inference* (Honkela et al., 2010), where the variational family $\mathcal{F}$ is restricted to parametric densities $q_\lambda(z|x)$ with parameters $\lambda \in \Lambda$. For instance, a Gaussian family with $q_\lambda(z|x) = \mathcal{N}(z; \boldsymbol{\mu}, \boldsymbol{\Sigma})$ corresponds to $\lambda = \{\boldsymbol{\mu}, \boldsymbol{\Sigma}\}$. Then, the E-step reduces to

$$\lambda^* = \arg\max_\lambda \mathcal{L}(\theta, \lambda; x). \tag{21}$$

For a dataset $\mathcal{X} = \{x_i\}_{i=1}^N$, the total ELBO becomes

$$\mathcal{L}(\theta, \lambda; \mathcal{X}) = \sum_{i=1}^N \mathcal{L}(\theta, \lambda_i; x_i), \tag{22}$$

but optimizing $N$ local parameters $\lambda_i$ quickly becomes unmanageable, especially in high dimensions. To overcome this, *amortized variational inference* assumes a parametric map $f_\phi$ such that $\lambda_i = f_\phi(x_i)$, making the ELBO

$$\mathcal{L}(\theta, \phi; \mathcal{X}) = \sum_{i=1}^N \mathbb{E}_{q_\phi(z_i|x_i)}\left[\log \rho_\theta(x_i, z_i) - \log q_\phi(z_i|x_i)\right]. \tag{23}$$

Thus, training the decoder $\rho_\theta(x|z)$ reduces to maximizing this objective jointly over $\theta$ and $\phi$, where $q_\phi(z|x)$ approximates the intractable posterior $\rho_\theta(z|x)$. The Variational Autoencoder (VAE) is a special case of this framework, where $q_\phi(z|x)$ is parameterized by a neural network (encoder). A common choice is Gaussian:

$$q_\phi(z|x) = \mathcal{N}(z; \boldsymbol{\mu}_\phi(x), \mathrm{diag}(\boldsymbol{\sigma}_\phi^2(x))), \tag{24}$$

with $\boldsymbol{\mu}_\phi$ and $\boldsymbol{\sigma}_\phi$ learned via backpropagation. Joint training of encoder and decoder can be suboptimal (He et al., 2018), yet it remains common.

Using KL divergence and Bayes' rule, we rewrite equation 23 as

$$\mathcal{L}(\theta, \phi; \mathcal{X}) = \sum_{i=1}^N \mathbb{E}_{q_\phi(z_i|x_i)}\left[\log \rho_\theta(x_i|z_i)\right] - \sum_{i=1}^N D_{\mathrm{KL}}\left[q_\phi(z_i|x_i) \| \rho(z)\right], \tag{25}$$

where the *first term* promotes reconstruction fidelity and the *second* regularizes the posterior to align with a prior (typically standard Gaussian), aiming for disentangled latent codes.

The last important point is that in computing gradients, we must avoid the intractability of the marginal likelihood. While the KL term is analytically tractable for Gaussians, the expectation term requires Monte Carlo sampling $z_i^{(r)} \sim q_\phi(z_i|x_i)$, but this sampling is non-differentiable. The reparametrization trick resolves this:

$$z_i^{(r)} = \boldsymbol{\mu}_\phi(x_i) + \mathrm{diag}(\boldsymbol{\sigma}_\phi^2(x_i))^{1/2}\epsilon^{(r)}, \quad \epsilon^{(r)} \sim \mathcal{N}(0, I), \tag{26}$$

enabling gradient flow through stochastic nodes. The resulting estimator is

$$\hat{\mathcal{L}}(\theta, \phi; \mathcal{X}) = \sum_{i=1}^{N} \frac{1}{R} \sum_{r=1}^{R} \log \rho_\theta(x_i|z_i^{(r)}) - \sum_{i=1}^{N} D_{\mathrm{KL}}\left[q_\phi(z_i|x_i)\|\rho(z)\right]. \tag{27}$$

VAEs thus optimize a surrogate for the log-likelihood while enabling efficient sampling from $\rho(z)$.

For completeness, we just mention some references showing main limitations of VAEs (Wei et al., 2020; Oussidi & Elhassouny, 2018; Sengupta et al., 2020): performance is sensitive to hyperparameters (e.g., prior choice, weightings), the latent structure may fail to disentangle data factors, and expressiveness may be limited due to simple priors. Generated samples may appear blurry or suffer from mode collapse, and VAEs often underperform on complex, high-dimensional datasets compared to alternative generative models.

### 3.2 Generative Adversarial Networks

Generative adversarial networks(Goodfellow et al., 2014) (GANs) are a class of generative models which take inspiration from game theory. They consist of two neural networks (see Figure 3), namely a *generator* G and a *discriminator* D, trained simultaneously through the so-called adversarial training. Given a dataset $\mathcal{X}$ sampled from the unknown data distribution $\rho_*$, the generator is devoted to generate synthetic data that ideally resembles the training data. On the other hand, the discriminator has to discern between fake and true samples. In this sense, G and D are adversary: the generator aims to produce realistic data to fool the discriminator, while the discriminator strives to correctly classify real and fake data. Thus, the training ends when the discriminator becomes unable to effectively distinguish between real and generated samples. Let us present the mathematical formulation: firstly we define a prior $\rho_z(z)$, which is a PDF easy to sample

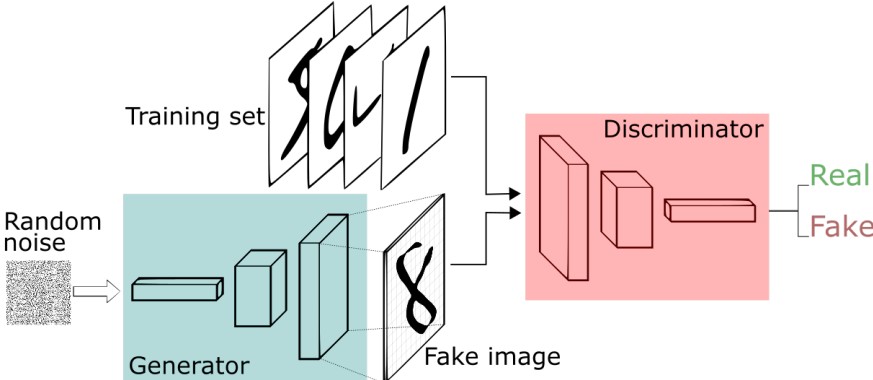

Figure 3: Scheme of the structure of GANs, taken from `https://sthalles.github.io/intro-to-gans/`.

from that serve to inject noise into the generator. The latter is a function $G_{\theta_g}(z)$ that is fed with noise and generate "fake" samples that should be similar to samples from $\rho_*$. The discriminator is a parametric function $D_{\theta_d}(x)$ that gives the probability that a sample $x$ comes from the training set rather than have been generated by $G$. Both $\theta_g$ and $\theta_d$ are parameters of a NN. The optimal weights are solution of the following two-player minimax problem:

$$\arg\min_{\theta_g} \arg\max_{\theta_d} \mathbb{E}_*[\log D_{\theta_d}(x)] + \mathbb{E}_{\rho_z}[\log(1 - D_{\theta_d}(G_{\theta_g}(z)))] := \arg\min_{\theta_g} \arg\max_{\theta_d} V(G, D) \tag{28}$$

We refer in the following to $\rho_g(x)$ as the distribution of "fake" samples induced by the generator, that is such that

$$\mathbb{E}_{\rho_z}[\log(1 - D_{\theta_d}(G_{\theta_g}(z)))] = \mathbb{E}_{\rho_g}[\log(1 - D_{\theta_d}(x))] \tag{29}$$

The empirical idea to solve the minimax game is via an alternating algorithm:

**Proposition 3.1.** *The optimization algorithm for a GAN is made by two alternating steps:*

- **Update of the discriminator**

  1. *Sample $\{z^{(i)}\}_{i=1}^N$ (noise) from $\rho_z$ and $\{x^{(i)}\}_{i=1}^N$ (data) from $\rho_*$.*
  2. *Compute $\nabla_{\theta_d} V(G, D)$ and perform* **gradient ascent** *to update $\theta_d$.*

- **Update of the generator**

  1. *Sample $\{z^{(i)}\}_{i=1}^N$ (noise) from $\rho_z$.*
  2. *Compute $\nabla_{\theta_g} V(G, D)$ and perform* **gradient descent** *to update $\theta_g$.*

This proposal is driven by common sense, but a more careful analysis of the minimax game is necessary to ensure convergence of such algorithm. In order to characterize the solutions of this adversarial game, it is necessary to search for the optima. The method of proof is: (1) classify solutions of optimization of $D$ at fixed $G$ and viceversa and then (2) present a convergence result of the alternating game. Let us start from the update of the discriminator:

**Theorem 3.1** (Existence of optimal discriminator(Goodfellow et al., 2014))**.** *For G fixed, the optimal discriminator D is*

$$D_G^* = \frac{\rho_*(x)}{\rho_*(x) + \rho_g(x)} \tag{30}$$

*Proof.* Using equation 28 and equation 29, we have

$$V(G, D) = \int_\Omega (\rho_*(x) \log D_{\theta_d}(x) + \rho_g(x)(1 - D_{\theta_d}(x))) dx \tag{31}$$

The function $y \to a \log(y) + b \log(1 - y)$ achieve its maximum in $(0, 1)$ at $a/(a+b)$ for $(a, b) \neq (0, 0)$. Applied to the case in study, the discriminator can be defined just in $Supp(\rho_*(x)) \cup Supp(\rho_g(x))$, hence concluding the proof. $\square$

This lemma ensure that the gradient ascending will eventually reach a maximum, that is

$$C(G) = \arg\max_D V(G, D) = \mathbb{E}_* \left[ \frac{\rho_*(x)}{\rho_*(x) + \rho_g(x)} \right] + \mathbb{E}_{\rho_g} \left[ \frac{\rho_g(x)}{\rho_*(x) + \rho_g(x)} \right] \tag{32}$$

Now we need to characterize the solutions of the minimization problem $\arg\min_G C(G)$

**Theorem 3.2** (Existence of optimal generator(Goodfellow et al., 2014))**.** *At fixed $D = D_G^*$, the optimal generator $G^*$ induce a $\rho_g$ such that $\rho_g = \rho_*$. At that point, $C(G^*) = -\log 4$.*

*Proof.* Regarding the last point, for $\rho_g = \rho_*$ we obtain $D_G^* = 1/2$, that inserted in $C(G)$ gives exactly $-\log 4$. We need to test whether this is a global optimum: we can sum and subtract $-\log 4$ to $C(G)$ obtaining

$$C(G) = -\log(4) + D_{KL}\left(\rho_* \,\middle\|\, \frac{\rho_* + \rho_g}{2}\right) + D_{KL}\left(\rho_g \,\middle\|\, \frac{\rho_* + \rho_g}{2}\right)$$
$$= -\log(4) + 2 \cdot JSD(\rho_* \,\|\, \rho_g) \tag{33}$$

where $JSD$ is the Jensen-Shannon divergence(Manning & Schutze, 1999). Such quantity has the same non-negativity property of KL divergence, i.e. $JSD(\rho_* \,\|\, \rho_g) \geq 0$ and $JSD(\rho_* \,\|\, \rho_g) = 0$ iff $\rho_g = \rho_*$. This proves that $\rho_g = \rho_*$, or more precisely the corresponding generator $G^*$, is the global minimum for $C(G)$. $\square$

To summarize, we have showed separate theoretical guarantees about convergence of gradient ascent and descent. However, we need to show that alternating those two steps would eventually converge to the global Nash equilibrium of the minimax game, i.e. $\rho_* = \rho_g$. The result is summarized in the following Theorem(Goodfellow et al., 2014) of which omit the proof for the sake of brevity.

**Theorem 3.3.** *If G and D have enough capacity, and at each step of the alternating algorithm, the discriminator is allowed to reach its optimum given G, and $\rho_g$ is updated so as to improve the criterion*

$$\mathbb{E}_*[\log D_G^*(x)] + \mathbb{E}_{\rho_g}[\log(1 - D_G^*(x)] \tag{34}$$

*then $\rho_g$ converges to $\rho_*$.*

Ideally, the theoretical treatment of Generative Adversarial Networks concludes with the proof that the proposed minimax game has a unique Nash equilibrium. This equilibrium corresponds to a generator capable of sampling from $\rho_*$, making it indistinguishable from true samples by the discriminator, performing no better than a random classifier with a probability of 1/2.

GANs have several drawbacks (Radford et al., 2016; Salimans et al., 2016; Arjovsky & Bottou, 2016), as highlighted by practical applications of Theorem 3.3. Optimization in parameter space on $\theta_g$ rather than functional space on $\rho_g$ introduces challenges, making what should be a convex problem non-convex. Additionally, training the generator is difficult due to gradients being near zero early on when the generator performs poorly. GANs are also known for training instability, requiring careful balancing between the generator and discriminator. Hyperparameter tuning is often needed, and GANs require large datasets for generalization. Sample generation can suffer from mode collapse, where outputs lack diversity, and computationally, GANs can be demanding, especially with high-resolution images. Evaluating GAN performance is difficult, as metrics like Inception Score and Frechet Inception Distance have limitations, complicating model comparison. Despite these issues, GANs remain important for unsupervised learning and model robustness.

### 3.3 Diffusion Models

Diffusion generative models (Sohl-Dickstein et al., 2015; Yang et al., 2023; Croitoru et al., 2023) are a class of models that use diffusion processes to transform a simple distribution into a more complex one over time. The process starts with a basic distribution, such as Gaussian noise, and iteratively transforms it into a target distribution that approximates real data, such as images. Recently, diffusion models have become state-of-the-art in several domains, partly replacing GANs (Dhariwal & Nichol, 2021). In this section, we summarize the common features of diffusion models without diving too deeply into specific variations.

As with other generative models, the core ingredient is a dataset $\mathcal{X} = x_{i\,i=1}^N$, where $x_i$ are drawn from an unknown target density $\rho_*(x)$. For simplicity, we assume $\mathcal{X} \subset \mathbb{R}^d$. In VAEs and GANs, the goal is to generate new samples from noise: VAEs decode from a Gaussian in latent space, and GANs generate from noise through the generator $G$. Similarly, in diffusion models, the aim is to push samples extracted from a simple distribution, like a Gaussian, toward the data distribution.

Since the following discussion is closely related to stochastic calculus (Rogers & Williams, 2000), we introduce the notation. We denote $X_t \in \mathbb{R}^d$ as a stochastic process, a sequence of random variables where $t \in \mathbb{R}$ represents continuous time. Unlike deterministic processes, the focus here is on the distribution in law of $X_t$, denoted $\rho(x,t)$, rather than on the individual trajectory. Just as deterministic trajectories are solutions to ordinary differential equations (ODEs), stochastic processes are solutions to stochastic differential equations (SDEs).

**Proposition 3.2** (SDE and Fokker-Plank PDE). *Given the drift $\mu : \mathbb{R}^d \times \mathbb{R} \to \mathbb{R}^d$ and the diffusion matrix $\sigma : \mathbb{R}^d \times \mathbb{R} \to \mathbb{R}^{d,d}$, let us consider the stochastic process $X_t$ solution for $t \in [0,T] \subset [0,+\infty]$ of the SDE*

$$dX_t = \mu(X_t,t)dt + \sigma(X_t,t)dW_t, \qquad X_0 \sim \rho_0 \tag{35}$$

*where $W_t$ is a Wiener process. Using Ito convention, the law of $X_t$, namely $\rho(x,t)$, satisfies the Fokker-Planck partial differential equation (PDE)*

$$\frac{\partial}{\partial t}\rho(x,t) = -\nabla \cdot [\mu(x,t)\rho(x,t)] + \Delta \left[\frac{\sigma(x,t)^2}{2}\rho(x,t)\right], \qquad \rho(x,0) = \rho_0(x) \tag{36}$$

This proposition is important to understand the relation between the single random process $X_t$ and its distribution in law. Let us present a simple example to clarify such connection.

**Example 3.1** (Wiener process). *Let us consider the case $\mu(x,t) = 0$ and $\sigma(x,t) = 1$ in $d = 1$, that corresponds to the SDE*

$$dX_t = dW_t \tag{37}$$

*The solution of the associated Fokker-Planck equation*

$$\frac{\partial \rho(x,t)}{\partial t} = \frac{1}{2}\frac{\partial^2 \rho(x,t)}{\partial x^2}, \tag{38}$$

*for a delta initial datum $\rho(x,0) = \delta(x)$ is precisely*

$$\rho(x,t) = \frac{1}{\sqrt{2\pi t}}e^{-x^2/2t} \tag{39}$$

*This is a gaussian density with variance proportional to t. That is, the initial concentrated density spreads on the real line.*

This brief summary about SDEs is sufficient to provide a consistent definition of generative diffusion model:

**Definition 3.2** (Generative diffusion model). *Let us consider the data distribution $\rho_* : \mathbb{R}^d \to \mathbb{R}_+$ and a base distribution $\bar{\rho}(x) : \mathbb{R}^d \to \mathbb{R}_+$. Given a time interval $[0,T] \in [0,\infty]$, a generative diffusion model is an SDE with fixed terminal condition*

$$dX_t = \mu(X_t,t)dt + \sigma(X_t,t)dW_t, \qquad X_0 \sim \bar{\rho}, \quad X_T \sim \rho_* \tag{40}$$

*where $W_t$ is a Wiener process.*

This definition resembles concepts from stochastic optimal control(Fleming & Rishel, 2012): in fact, the terminal condition is not sufficient to uniquely fix $\mu(X_t,t)$ and $\sigma(X_t,t)$. Under this point of view, the specification of a particular class of diffusion models reduces to a *prescription on how to determine the drift and the diffusion matrix*. In the following we will summarize two highlighted methods present in literature.

**Score-based diffusion(Song et al., 2020b).** To explain what is score-based diffusion we need the following preliminaries:

**Definition 3.3.** *Given a PDF $\rho(x)$, the **score** is the vector field*

$$s(x) = \nabla \log \rho(x) \tag{41}$$

**Proposition 3.3** (Naive score-based diffusion). *For any $\varepsilon > 0$ and $\rho_0(x)$, the choice $\mu(x,t) = \varepsilon s_*(x) = \varepsilon \nabla \log \rho_*(x)$ and $\sigma = \sqrt{2\varepsilon}$ in equation 40 satisfies the endpoint condition for $T = \infty$.*

*Proof.* If we consider the Fokker-Planck PDE associated to equation 40 with the selected drift and variance, we have

$$\partial_t \rho(x,t) = \nabla \cdot [-s_*(x)\rho(x,t) + \nabla\rho(x,t)] = \nabla \cdot \left[\rho(x,t)\nabla \log\left(\frac{\rho(x,t)}{\rho_*(x)}\right)\right] \tag{42}$$

By direct substitution, the stationary probability density $\rho_*(x)$ is a solution. For uniqueness, we need to prove that any solution of the PDE would converge to this solution. A formal argument is based on Jordan-Kinderlehrer-Otto (JKO) variational formulation of Fokker-Planck equation(Jordan et al., 1998), interpreted as a gradient flow in probability space with respect to Wasserstein-2 distance. An alternative way is the following: for any solution $\rho(x,t)$, we can compute the time derivative of the KL divergence between such solution and $\rho_*(x)$. If we define $R = \rho/\rho_*$:

$$\frac{d}{dt}D_{\mathrm{KL}}(\rho \parallel \rho_*) = \frac{d}{dt}\int_{\mathbb{R}^d} \rho \log R \, dx = \int_{\mathbb{R}^d} \partial_t \rho \log R \, dx + \int_{\mathbb{R}^d} \frac{\rho}{R}\partial_t R \, dx \tag{43}$$

We can use Fokker-Planck equation to substitute $\partial_t \rho$ and integrate by parts:

$$\frac{d}{dt} D_{\mathrm{KL}}(\rho \parallel \rho_*) = \frac{d}{dt} \int_{\mathbb{R}^d} \rho \log R \, dx = -\int_{\mathbb{R}^d} \rho |\nabla \log R|^2 \, dx + \int_{\mathbb{R}^d} \rho_* \partial_t R \, dx \tag{44}$$

We notice that $\rho_* \partial_t R = \nabla \cdot (\rho \log R)$, hence that the second addend is zero by integration by parts. The conclusion is that

$$\frac{d}{dt} D_{\mathrm{KL}}(\rho \parallel \rho_*) = -\int_{\mathbb{R}^d} \rho |\nabla \log R|^2 \, dx \le 0, \tag{45}$$

concluding the proof. □

The result seems to say that we are able to build a diffusion generative models estimating the score of the target. In a data driven context, $\rho_*$ is known just through data points and one has to face the problem of estimating $s_*$. A possible approach(Hyvärinen & Dayan, 2005) is score matching.

**Definition 3.4** (Fisher divergence). *Given two PDFs $\rho(x)$ and $\pi(x)$, the **Fisher divergence** is defined as*

$$D_F(\rho \parallel \pi) = \int_{\mathbb{R}^d} \rho(x) \|\nabla \log \rho(x) - \nabla \log \pi(x)\|^2 \, dx \tag{46}$$

Even if in some sense $D_F$ seems to measure some distance between two PDFs, it is very different from the KL divergence, see following Remark.

**Remark 3.1.** *By definition, both KL and Fisher divergence between two PDFs satisfy the non-negativity property, i.e. they are strictly positive, and zero only when the densities are the same. $D_F$ does not depend on normalization constants of the PDFs because of the gradients. This is a double-edged weapon: it is apparently useful in high dimension, where the computation of normalization of a density is impractical (as for instance the partition function for EBMs). But if the distribution is multimodal, the local nature of $D_F$ is very insensible to global characteristics of the densities, as for instance the relative mass in each mode. Let us consider a key example: the distributions we would like to compare are:*

$$\begin{aligned} \rho_1(x) &= 0.5\mathcal{N}(x, -5, 1)(x) + 0.5\mathcal{N}(x, 5, 1), \\ \rho_2(x) &= \sigma(z)\mathcal{N}(x, -5, 1) + (1 - \sigma(z))\mathcal{N}(x, 5, 1) \end{aligned} \tag{47}$$

*where $\sigma(z) = 1/(1 + e^{-z})$ is a sigmoid function. The two densities are bimodal gaussian mixture in 1D with same means and variances; the second mixture is balanced with relative mass equal to $1/2$. We would like to compare $D_F(\rho_1 \parallel \rho_2)$ and $D_{KL}(\rho_1 \parallel \rho_2)$ as functions of $z$. In Figure 4 we plot the two divergences in function of $z$. We estimate the expectations that define the two divergences using a Monte Carlo estimate, namely*

$$D_F(\rho_1 \parallel \rho_2) \approx \sum_{i=1}^{N} \|\nabla \log \rho_1(x_i) - \nabla \log \rho_2(x_i)\|^2$$

$$D_{KL}(\rho_1 \parallel \rho_2) \approx \sum_{i=1}^{N} \log \left[ \frac{\rho_1(x_i)}{\rho_2(x_i)} \right] \tag{48}$$

*where $x_i \sim \rho_1(x)$. The minimum value is 0 and corresponds to $z = 0$, that is $\rho_1 = \rho_2$. The first difference is that the values of $D_F$ are smaller of several order of magnitude — in general, this could be a problem in practical implementations. Most importantly, the shape of the curve is very different. In this one dimensional example we need $N = 10000$ to appreciate a similar growth, even if $D_F$ curve is more steep. For smaller $N$, $D_F$ is basically flat for $z \ne 0$. This is related to the absence of points in low density regions, that is where the integrand in $D_F$ gives a non-zero contribution.*

In score matching, one propose a parametric score $s_\theta$, for instance a neural network, and train such model to match the true score $s_*$. The loss on which the model is trained, using for instance gradient routines, is

$$\mathcal{L}(s_\theta, \rho_*) = \frac{1}{2} \int_{\mathbb{R}^d} \rho_* \|s_\theta - \nabla \log \rho_*\|^2 \, dx = \mathbb{E}_*[\|s_\theta\|^2 + \mathrm{tr}(J_x s_\theta)] + C_p \tag{49}$$

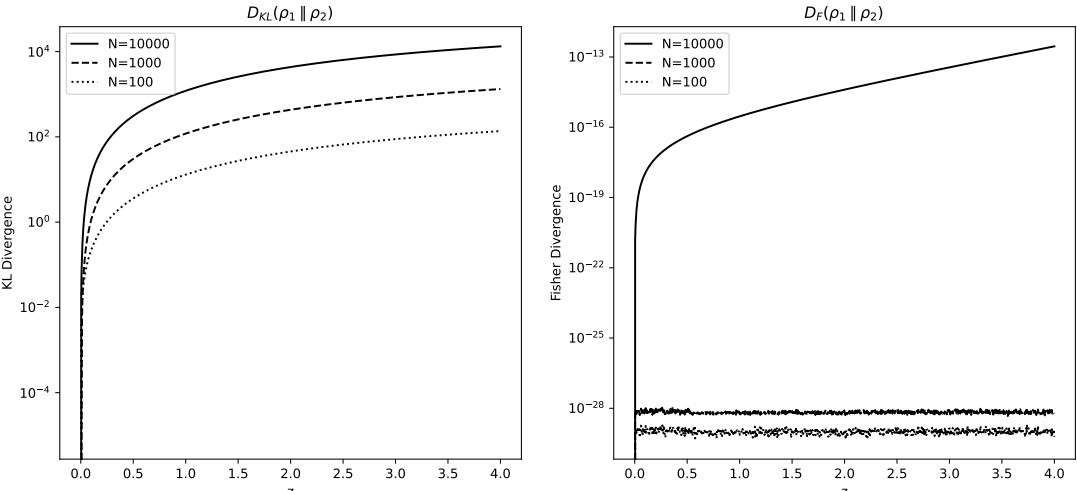

Figure 4: Comparison between KL divergence and Fisher divergence for the two bimodal gaussian mixtures in equation 47. The variable $z \in (0, \infty)$ is related to the relative mass of the two modes via a sigmoid function $\sigma(z) \in (0, 1)$; the plots for $z < 0$ are analogous by symmetry. Notice the different scales of the $y$ axes. The Monte Carlo estimation is performed using $N = 100, 1000, 10000$ samples.

where we integrated by parts, $C_p = const$ does not depend on $\theta$ and $J_x$ is the Jacobian with respect to $x$. Denoting with $\rho_\theta$ one (n.b. not unique) PDF associated to $s_\theta$, the loss is evidently $D_F(\rho_* \parallel \rho_\theta)$. The right hand side reformulation in equation 49 is crucial: the expectation $\mathbb{E}_*$ can be estimated using data points at our disposal, bypassing the problematic term $\nabla \log \rho_*$.

Unfortunately, the naive score-based approach is plagued by two fundamental issues that make it impractical. The first regards the score estimation itself: usually, data at our disposal comes from high density region of $\rho_*$, that is the estimation of $\mathbb{E}_*$, hence of the score, will be inaccurate outside such areas. The problem is that the initial condition (e.g. noise) of the SDE is usually located far from data. An imprecise drift will critically affect the generation process, leading to unpredictable outcomes. The second regards the difference between the PDE and the practical implementation through equation 40. The generation problem is convex in probability space, i.e. $\rho_*$ is the unique asymptotic stationary solution, but the rate of convergence of the law of $X_t$ is critically related to the particular $\rho_*$ in study, in particular in relation with multimodality and slow mixing. We will discuss in details about this issue in Section 4.

The next step towards state-of-the-art score-based diffusion is the following lemma(Anderson, 1982):

**Lemma 3.2.** *Any SDE in the form*

$$dX_t = f(X_t, t)dt + g(t)dW_t, \qquad X_0 \sim \rho_1, \quad X_T \sim \rho_2 \tag{50}$$

*with solution $X_t \sim \rho(x, t)$ admits an associated reversed SDE*

$$dX_s = [f(X_s, s) - g(s)^2 \nabla \log \rho(x, s)]ds + g(s)dW_s, \qquad X_T \sim \rho_2, \quad X_0 \sim \rho_1 \tag{51}$$

*where $ds$ is a negative infinitesimal time step and $s$ flows backward from $T$ to $0$. By convention, equation 50 is also called forward SDE and equation 51 backward one.*

Exploiting this result, we can define a score-based diffusion model:

**Definition 3.5.** *A* **score-based generative model** *is the backward SDE equation 51, where $\rho_1 = \rho_*$ and $\rho_2 = \bar{\rho}$.*

Apparently, the situation is even worse with respect to score matching: the score in equation 51 is related to the law of $X_t$, i.e. it is time dependent and generally not analytically known — score estimation was already

an issue for $s_*(x)$. The core idea in score-based diffusion is to extract information about $\nabla \log \rho(x, s)$ from the forward process since the solutions of equation 50 and equation 51 have the same law, see Figure 5. By

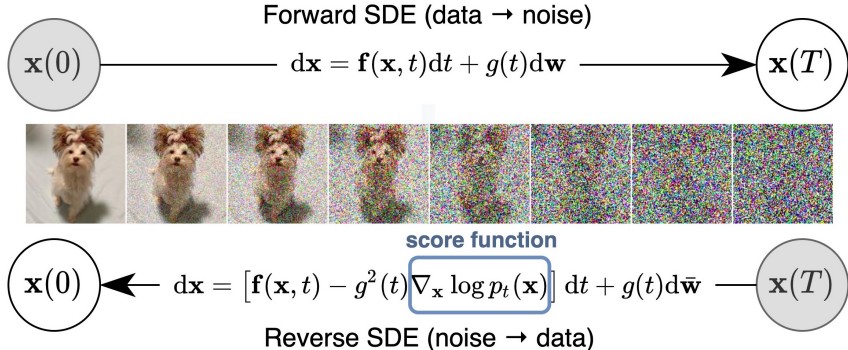

Figure 5: Schematic representation of forward and backward process in score-based diffusion. Image taken from (Song et al., 2020b).

Definition 3.5, the forward process brings *data to noise* and the model to be learned is a time dependent parametric vector field $s_\theta(x, t)$. The loss used during the forward process to learn the score is:

$$\mathcal{L}_{SM}(s_\theta(x, t)) = \mathbb{E}_{t \sim \mathcal{U}(0,T)} \mathbb{E}_{\rho(x,t)}[\lambda(t) \| s_\theta(x, t) - \nabla \log \rho(x, t) \|^2] \tag{52}$$

where $\lambda : [0, T] \to \mathbb{R}_+$ is a positive scalar weight function and $U(0, T)$ is the uniform distribution in $(0, T)$. After the same integration by parts used in equation 52, there is still the problem that computing the hessian is expensive in high dimension, especially if $s_\theta$ is a neural network. Several proposal to solve this issue have been proposed and successfully exploited, such as denoising score matching(Vincent, 2011) or sliced score matching(Song et al., 2020a). Another subtle issue is that generally the forward process would generate pure noise for $T = \infty$ — one could be worried that the truncation at finite time would provide an imprecise estimate of the score at that time scale, that is close to noise, and induce errors during the generative phase. This problem is attacked by practitioners via several tricks but the theoretical results in this sense are not complete.

Let us provide a brief interpretation of why score-based diffusion works better than naive score matching (Proposition 3.3). Let us consider the simple case $f(x, t) = 0$ and $g(t) = e^t$; the resulting forward process is perturbing data with gaussian noise at increasing variance scale(Song & Kingma, 2021). That is, time scale corresponds to amount of noise in this setup. We recall that the problem of naive score matching was the lack of data in low density region for the target density. In score-based diffusion one use perturbed data to populate those region and compute the score at each time scale that serves as bridge from $\bar{\rho}$ and the target $\rho^*$. A more recent class of diffusion-based generative models is the framework of stochastic interpolants (Albergo & Vanden-Eijnden, 2022), offering a unified view through continuous stochastic processes. This approach is related to flow-based methods, such as rectified flow and flow matching (Liu et al., 2022; Lipman et al., 2022), which can be seen as different parameterizations of similar principles. These methods transform a simple prior distribution (e.g., Gaussian noise) into a complex target distribution by modeling probability flow dynamics. With strong theoretical foundations and empirical success, they form the basis of state-of-the-art generative models like Stable Diffusion 3[2]. We present in Appendix B a brief review of stochastic interpolation, even if not strictly related to EBMs. Moreover, it is worth mentioning as flow-based generative models are similar to the Schrödinger Bridge method (De Bortoli et al., 2021), which solves an entropy-regularized optimal transport problem. Unlike flow-based models, Schrödinger Bridges match distributions while minimizing transport cost under stochastic constraints, making them more complex to solve as they require iteratively solving coupled forward and backward processes bridging target and initial densities.

---

[2]https://encord.com/blog/stable-diffusion-3-text-to-image-model/

### 3.4 Normalizing Flows

The fundamental idea underlying Normalizing Flows(Tabak & Vanden-Eijnden, 2010; Tabak & Turner, 2013) (NF) is very close to the usual in generative modelling: to transform samples from a straightforward base distribution, often a Gaussian, to data distribution. The main feature of NF is that the transformation is performed through a series of invertible and differentiable transformations.

The core concept revolves around constructing a model capable of learning a sequence of *invertible* operations that can map samples from a simple distribution to the target distribution. In particular, we recall the well-known lemma:

**Lemma 3.3.** *Let us consider a random variable $Z \in \mathbb{R}^d$ and its associated probability density function $\rho_Z(z)$. Given an invertible function $Y = \phi(Z)$ on $\mathbb{R}^d$, the probability density function in the variable $Y$ is defined through*

$$\rho_Y(y) = \rho_Z(g^{-1}(y))|\det J_y\varphi^{-1}(y)| = \rho_Z(\phi^{-1}(y))|\det J_y\phi(\phi^{-1}(y))|^{-1} \tag{53}$$

*where $\phi^{-1}$ is the inverse of $\phi$ and $J_y$ is the Jacobian w.r.t. $y$. The density $\rho_Y$ is also called **pushforward** of $\rho_Z$ by the function $\phi$ and denoted by $\phi_\#\rho_Z$.*

In generative modelling, $\rho_Z$ is identified with the base distribution and its pushforward as the target, i.e. data, distribution. The direction from noise to data is called generative direction, while the other way is called normalizing direction — data are normalized, gaussianized, by the inverse of $\phi$. The name Normalizing Flow originates from the latter. In fact, the mathematical foundation of NF is reduced to Lemma 3.3.

The whole problem reduces to design the pushforward in a data driven setup, that is where we just have a dataset $\mathcal{X}$ of samples from the target and no access to the analytic form of $\rho_*$. In order to link NF with other generative models, let us denote with $\phi_\theta$ with $\theta \in \Theta$ the parametric map that characterizes the pushforward $\rho_\theta = (\phi_\theta)_\#\rho_Z$. In practice, this map is usually a neural network and $\rho_\theta$ will implicitly depend on it. The optimal parameters $\theta^*$ are chosen to be solution of the following optimization problem:

$$\theta^* = \arg\min_{\theta\in\Theta} D_{\mathrm{KL}}(\rho_*(x) \parallel \rho_\theta(x)) = \arg\max_{\theta\in\Theta} \mathbb{E}_*[\log \rho_\theta(x)] \tag{54}$$

As already stressed, this formulation in term of maximum log-likelihood is equivalent to cross-entropy minimization for EBMs. As for VAEs, the analytical form of $\rho_\theta$ is not known: in NF it is implicitly defined through the pushforward. This issue is attacked using Lemma 3.3 to rewrite the right hand side in equation 54 as

$$\arg\max_{\theta\in\Theta} \mathbb{E}_*[\log \rho_\theta(x)] = \arg\max_{\theta\in\Theta} \mathbb{E}_*[\log \rho_Z(\phi_\theta^{-1}(y)) + \log|\det J_y\phi^{-1}(y)|] \tag{55}$$

The likelihood of a sample under the base measure is represented as the first term, and the second term, often referred to as the log-determinant or volume correction, accommodates the alteration in volume resulting from the transformation introduced by the normalizing flows. After this manipulation every addend inside the expectation is calculable — the map $\phi$ and the noise distribution $\rho_Z$ are given (e.g. a gaussian). As usual, the expectation can estimated via Monte Carlo using the finite dataset $\mathcal{X}$ at our disposal. Any gradient based optimization routine can be then exploited to optimize $\theta$. During training, the model adjusts the parameters $\theta$ to bring the transformed distribution in close alignment with the true data distribution.

The main limitation in NF is that the pushforward map must be bijective for any $\theta$. Not only that: both forward and inverse operations are required to be computationally feasible to perform generation and normalization. Furthermore, the Jacobian determinant must be tractable to facilitate efficient computation. These requests constrain the possible neural architectures that one can use to model $\phi_\theta$. The following lemma provides a decisive tool in this sense.

**Lemma 3.4.** *Let us consider a set of $M$ bijective functions $\{f_i\}_{i=1}^M$. If we denote with $f = f_M \circ f_{M-1} \circ \cdots \circ f_1$ their composition, one can prove that $f$ is bijective and its inverse is*

$$f^{-1} = f_1^{-1} \circ \cdots \circ f_{M-1}^{-1} \circ f_M^{-1} \tag{56}$$

*Moreover, if we denote with $x_i = f_i \circ \cdots \circ f_1(z) = f_{i+1}^{-1} \circ \cdots \circ f_M^{-1}$ and $y = x_M$, we have*

$$\det J_y f^{-1}(y) = \prod_{i=1}^M \det J_y f_i^{-1}(x_i) \tag{57}$$

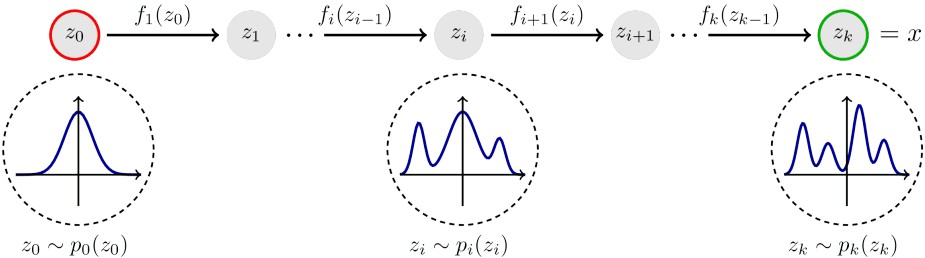

Figure 6: Schematic representation of Normalizing Flows, image taken from `https://flowtorch.ai/users/`.

| | Implementation of Maximum Log-Likelihood |
|---|---|
| EBM | Cross-Entropy Minimization
$\arg\min_{\theta \in \Theta} \mathbb{E}_* [U_\theta] + \log Z_\theta$ |
| VAE | Latent Space
$\arg\max_{\theta, \theta'} \sum_{i=1}^{N} \frac{1}{R} \sum_{i=r}^{R} \log \rho_\theta(x_i | z_i^{(r)}) - \sum_{i=1}^{N} D_{\mathrm{KL}}[\log q_{\theta'}(z_i | x_i) \| \rho(z_i)]$ |
| GAN | Minimax Game
$\arg\min_{\theta} \arg\max_{\theta'} \mathbb{E}_* [\log D_{\theta'}(x)] + \mathbb{E}_{\rho_z}[\log(1 - D_{\theta'}(G_\theta(z)))]$ |
| SBD or SI | Implicit via Transport-Diffusion Equation
$\partial_t \rho(x, t) + \nabla \cdot ((b_\theta(x, t) + \varepsilon s_{\theta'}(x, t))\rho(x, t) - \varepsilon \nabla \rho(x, t)) = 0$ |
| NF | Volume Correction Factor
$\arg\max_{\theta \in \Theta} \mathbb{E}_* [\log \rho_Z(\phi_\theta^{-1}(y)) + \log |\det J_y \phi_\theta^{-1}(y)|]$ |

Table 1: Comparison of implementation of maximum log-likelihood for different generative models.

Exploiting this factorization result, the strategy is to compose invertible building blocks $(\phi_\theta)_i$ to construct a function $\phi_\theta$ that is sufficiently expressive. In general, the architecture of Normalizing Flows encompasses various transformations (see Figure 6), including simple operations like affine transformations and permutations, as well as more complex functions such as coupling layers. Common flow architectures include RealNVP, Glow, and Planar Flows, each introducing unique ways to parameterize and structure transformations(Kobyzev et al., 2020).

Regarding drawbacks of NF, one significant limitation lies in the computational cost associated with training NF, particularly as model complexity increases. The requirement for invertibility and the computation of determinants of Jacobian matrices contributes to the time-consuming nature of training, especially in deep architectures.

The architectural complexity of NF poses another challenge. Designing an optimal structure and tuning parameters may prove challenging, necessitating experimentation and careful consideration. Moreover, they may face challenges in scaling to extremely high-dimensional spaces, limiting their applicability in certain scenarios. Despite their expressiveness, NF may struggle to capture extremely complex distributions, requiring an impractical number of transformations to model certain intricate data distributions effectively.

Another degree of freedom is the choice of the base distribution $\rho_Z$, which can impact NF performance. Using a base distribution that does not align well with the true data distribution may hinder the model's ability to accurately capture underlying patterns. Training NF is observed to be less stable compared to other generative models, requiring careful tuning of hyperparameters and training strategies to achieve convergence and avoid issues like mode collapse. Lastly, interpreting the learned representations and transformations within NF can be challenging, which is an obstacle for a straightforward comprehension of how the model captures and represents information.

### 3.5 Comparison with EBMs

In this section, we present a summarized comparison of EBMs and the other generative models. First of all, the main similarity is the objective: maximize the log-likelihood is the general aim. In Table 1 we present how this task is instantiated case by case. In generative models, there exists an inherent trade-

off between the model's ability to generate data and its alignment with real-world data. Essentially, the paradigm followed in each optimization step involves two key stages: (1) the generation of data using a fixed model, and (2) the evaluation of the model's performance by comparing the generated ("fake") data with the actual dataset. This dual-step process is universally applicable, albeit with variations in implementation. It represents an interpretation of generative models as a balance between their discriminative and sampling capabilities. The key difference among generative models lies in how they learn. Normalizing Flows and Generative Adversarial Networks directly learn a deterministic mapping from data to noise, while VAEs, diffusion models, and EBMs rely on a stochastic sampling process. Deterministic methods can be more efficient but are prone to amplifying errors from finite datasets, whereas stochastic approaches, like Score-Based Diffusion, benefit from noise as a regularizer. The choice of noise level in diffusion models and latent structure in VAEs remains an open challenge. Most generative models are bidirectional, except for GANs, where invertibility is absent. Recent models, such as Score-Based Diffusion, improve fidelity by leveraging information from the noising process. This suggests that tools from Mathematical Physics, particularly stochastic processes, could provide deeper insights into generative modeling and drive future advancements.

Now, let's delve into a more detailed mathematical comparison, with a focus on Energy-Based Models . Specifically, we demonstrate how, in certain cases, other generative models can be interpreted as EBMs:

- For GANs, if the discriminator is $D_{\theta'}(x) \propto e^{-U_{\theta'}(x)}$, we immediately recover the term $\mathbb{E}_*[U_\theta]$. The training of the generator correspond to learn a perfect sampler, and resemble the use of machine learning to improve MCMC in computational science(Song et al., 2017).

- For SBD and SI, if the score is modelled by $s_{\theta'}(x,t) \propto -\nabla U_{\theta'}(x)$ the law of the process solution of the SDE is a Boltzmann-Gibbs ensemble by construction. Thus, the strong analogy is related to the constrained structure imposed to the law of the bridging process between the noise and the data, forced to be a BG ensemble. Regarding the loss, since the model is trained on Fisher divergence or on the interpolants, there is no direct analogy between the losses.

- For NFs, if $\phi_\theta$ is the map associated to a flow that brings $X_0 \sim \rho_Z(\phi_\theta^{-1}(x)) \propto e^{-U_\theta(x)}$ to $X_1 \sim \rho_*$, then the term $\mathbb{E}_*[U_\theta]$ present in EBMs is analogous to $\mathbb{E}_*[\log \rho_Z(\phi_\theta^{-1}(y))$ for NFs. In practice, if the composition of $\rho_Z$ with the normalizing flow can be written as an EBM, there is no difference between the models. This is of course not true in general — it is not given that for any $\theta$, a composition of the inverse map $\phi_\theta^{-1}$ and $\rho_Z$ can be always associated to an EBM parameterized via $U_\theta$.

- For VAEs the situation is a bit more intricate because of the ELBO reformulation. A possible interpretation towards EBMs is to think about encoder and decoder as a forward and backward processes from data to a latent space (possibly independent features, similarly to gaussian noise). One could imagine $\rho_\theta(x_i|z_i^{(r)})$ and $q_{\theta'}(z_i|x_i)$ as EBMs that have to match with some constraint on the $z$ space. In fact, the original EM steps represent an alternating optimization, where $\theta$ is not related to $\theta'$. In this sense, VAEs tries to match the forward and backward processes, similarly to SBD where they are the same by construction of the score.

When comparing the theoretical properties of generative models with EBMs, several aspects are worth to be stressed, such as **approximation properties**, **sample efficiency**, and the ability to handle **manifold data**.

- **Approximation Properties**: One of the key distinctions between models is their ability to approximate complex distributions. Normalizing Flows offer excellent approximation properties due to their ability to exactly model any distribution through a sequence of invertible transformations (Papamakarios et al., 2021). However, these transformations can become computationally expensive as the complexity of the data grows. On the other hand, EBMs also have strong approximation abilities but rely on the flexibility of the energy function. EBMs can approximate complex distributions well, but the necessity of calculating the partition function $Z$ and performing sampling (e.g., MCMC) makes this process more complex, especially for high-dimensional spaces.

- **Sample Efficiency**: The efficiency of generating high-quality samples differs among models. GANs are typically faster in generating samples compared to EBMs, which require iterative sampling methods such as MCMC. Diffusion models and flow-based methods, while offering impressive generation quality, can also be computationally expensive in practice, although recent advancements in efficient sampling techniques have improved this almost beyond GANs performance (Ma et al., 2022; Liu et al., 2022; Tong et al., 2023). NFs can generate samples quickly once the transformation is learned, but their scalability and sample efficiency degrade with the dimensionality of the data.

- **Ability to Handle Manifold Data**: Models like VAEs (Zheng et al., 2022) and NFs (Köhler et al., 2021) perform well on manifold data due to their explicit modeling of latent spaces and invertible mappings. EBMs, due to their flexible energy function, can also handle manifold data but require careful design of the energy function to ensure that the model appropriately reflects the underlying structure of the data (Arbel et al., 2021). Diffusion models naturally handle continuous data spaces and can be adapted to deal with manifold structures by carefully designing the noising and denoising processes (Pidstrigach, 2022).

In conclusion, generative models can be seen as bridges linking a simple source, like noise, to real data. Just as constructing a physical bridge requires stable foundations, building a generative model involves statistical analysis of the dataset and selecting an appropriate noise source. Data preprocessing, akin to adjusting docking configurations, is crucial for proper alignment. Different generative models suit different data structures, much like roads adapt to terrain, with the goal of maximizing log-likelihood to ensure an effective connection between source and target distributions.

## 3.6 Why (and why not) EBMs in practice

First of all, a key advantage of EBMs over other generative models is their ability to define an explicit probability density function in terms of an energy function. Unlike models that indirectly parameterize a distribution (e.g., VAEs, GANs, or diffusion models) or impose structural constraints to ensure tractability (e.g., normalizing flows), EBMs offer a more flexible framework in which the probability of a sample is directly related to its energy through the Boltzmann distribution. This explicit formulation, assuming a computable partition function Z along training (cfr. (Carbone et al., 2024a)), allows EBMs to provide normalized probabilities for arbitrary data points, making them particularly suitable for tasks such as anomaly detection (Zhai et al., 2016; Yoon et al., 2024), likelihood-based evaluation (Du & Mordatch, 2019), and probabilistic reasoning (LeCun et al., 2007). Unlike normalizing flows, which rely on invertible transformations, EBMs impose no such constraints, making them highly expressive. Moreover, they naturally generalize many probabilistic models used in physics and machine learning, providing a strong theoretical foundation that connects them to statistical mechanics and variational inference (Huembeli et al., 2022; Cheng & Courville, 2024). Because energy functions directly shape the probability landscape, practitioners can analyze how different regions of input space contribute to model confidence, uncovering structured relationships in the data, like manifold identification (Yang & Ji, 2023; Arbel et al., 2021) – this fact, together with the direct probabilistic content, makes Energy-Based Models more *interpretable* with respect to other generative models. This property is particularly valuable in scientific applications and settings where understanding model decisions is as important as predictive performance, for instance in molecular graph generation (Liu et al., 2021), molecular reactions (Sun et al., 2021), text modelling (Yu et al., 2022), agent policy regulation (Sharma et al., 2021), predictions of human behaviours (Pang et al., 2021), protein modelling (Frey et al., 2023), high-energy physics (Cheng & Courville, 2024), statistics (Zhang et al., 2022), drug discovery (Li et al., 2023), out-of-distribution detection (Liu et al., 2020), and many others.

While EBMs offer significant advantages in terms of explicit probabilistic modeling and interpretability, they may not be the ideal choice when efficient and fast sample generation is the primary goal. The need for methods like MCMC for sampling means EBMs can be computationally expensive and slow, particularly in high-dimensional spaces, see Section 4. In tasks where real-time generation is crucial—such as image synthesis, text generation, or interactive applications—models like GANs, diffusion models, or flow-based models are often preferred. These models are capable of producing high-quality samples in a much more efficient and faster manner, making them better suited for scenarios where speed and scalability are essential. Moreover, other generative models tend to outperform EBMs in terms of sample quality. For instance,

GANs are well-suited for applications that prioritize high-quality generative performance without the need for in-depth probabilistic insight.

There have been tentative hybrid approaches attempting to combine the strengths of EBMs with other generative models, such as diffusion-based models and flow-matching techniques (Du et al., 2023; Chao et al., 2024). These hybrid models aim to leverage the sampling efficiency and generation quality of diffusion models or normalizing flows, while also incorporating the probabilistic grounding and interpretability of EBMs. By combining the advantages of both paradigms, these approaches seek to obtain the best of both worlds—offering more efficient generation without sacrificing the explicit probabilistic framework provided by EBMs.

*Take Home Message*: EBMs are very useful and preferable for tasks that require **detailed probabilistic understanding** and **interpretability**. On the contrary, they are generally less practical for rapid, large-scale generation or when interpretability is not a major concern.

## 4  EBMs and sampling

The challenge of sampling from the Boltzmann-Gibbs (BG) ensemble arises in statistical mechanics, particularly when dealing with complex systems at equilibrium. This ensemble encapsulates the probability distribution of states for a system with numerous interacting particles at a given temperature. The primary obstacle in that context lies in the exponential number of possible states and intricate dependencies among particles, rendering brute-force methods impractical for large systems. A similar difficulty is encountered during EBM training, since the computation of the expectation $\mathbb{E}_\theta$ requires the ability to sample from a Boltzmann-Gibbs density.

Let us restrict to the case in which the energy $U(x)$ is defined on $\mathbb{R}^d$, that corresponds to continuous states in Statistical Physics. Any proposed techniques to efficiently sample from $\rho_{BG}(x) = \exp(-U(x))/Z$ can rely just on $U(x)$ or on its derivatives, even if the computation of many iterated derivatives can be expensive in high dimension. The estimation of the partition function or the shape of the energy landscape are in general unknown — on the contrary, they are the unknowns. Methods as rejecting sampling(Casella et al., 2004) cannot be used since one has usually access to $U(x)$ and not to the normalized density $\rho_{BG}(x)$. Since the advent of computational science, sampling has been attacked with many methods — a complete and exhaustive review of the existent methods would lead us off-topic. In this Section, we will highlight three common routines for sampling from a BG enseble: Metropolis-Hastings and Unadjusted Langevin Algorithm, and lastly Metropolis Adjusted Langevin Algorithm, a sort of fusion of the first two.

Let us better define the mathematical setting. We consider a space $\Omega \subseteq \mathbb{R}^d$ and a discrete sequence $(t_k)_{k\geq0} \subset \mathbb{N}$. Then, we consider $X_{t_k} \coloneqq X_k$ to be a stochastic process in $\Omega$ and discrete time. For the sake of simplicity we will always consider absolutely continuous densities with respect to Lebesgue measure.

**Definition 4.1** (Informal). *Sampling from a BG ensemble consists in defining the process $X_k$ such that $\exists T > 0$, not necessarily unique, for which $X_T \sim \exp(-U(x))/Z$.*

Once we manage to define such a process, and implement it in practice, we have solved the problem of sampling from a BG ensemble. A possible implicit way to define such stochastic process is via a transition kernel. Suppose we are interested in the law of the process $X$ at time $k + 1$, that we denote $\rho(X_{k+1})$ with an abuse of notation (n.b. analogous of $\rho(x, t)$ in the context of SDEs and Fokker-Planck equation). By definition of conditional probability, there exists a function $T : \Omega^{n+1} \to \mathbb{R}_+$ such that

$$\rho(X_{k+1}) = \int_{\Omega^d} T(X_{k+1}|X_k, \ldots, X_0)\rho(X_k, \ldots, X_0) \prod_{i=0}^{n} dX_i \tag{58}$$

This equation asserts that any property, uniquely defined by the law $\rho(X_{k+1})$ of the system at time $k + 1$, depends on the system's state at any $k \geq 0$. Generally, this strict constraint is relaxed by imposing Markovianity(Stroock, 2013), which is the property of the transition kernel to depend solely on the present state $X_k$ and not on previous states, i.e.

$$T(X_{k+1}|X_k, \ldots, X_0) = T(X_{k+1}|X_k) \coloneqq T(X_k, X_{k+1}) \tag{59}$$

The sequence $(X_k)n \geq 0$ is called a Markov chain if the associated transition kernel is Markovian. The question now is how to design such a chain to solve the sampling problem. Traditionally, it is simpler to identify a transition kernel for which $\rho_{BG}(x)$ is the unique stationary distribution, i.e., $\rho(X_k) = \rho_{BG}(x)$ for any $n > T$ in Definition 4.1. Moreover, the integral definition equation 58 is not suitable for applications since one usually evolves $X_k$ and not its law. Typically, it is required that $T$ is associated with an explicit time evolution for the process, namely an explicit mapping $X_{k+1} = F(X_k)$.

For historical reasons, let us present the most famous procedure to build the required sampling stochastic process, namely the Metropolis-Hastings algorithm(Metropolis et al., 1953; Hastings, 1970). Such techniques stand out as a foundational Markov chain Monte Carlo (MCMC)(Andrieu et al., 2003) method. Here, we provide its definition and a sketch of the proof of its properties.

**Definition 4.2** (Metropolis-Hastings (MH) algorithm). *Let us consider an initial condition $X_0 \sim \rho_0(x)$, where $\rho_0(x)$ simple to sample from (e.g. Gaussian or uniform). Let us consider a conditional probability distribution $g(X_{k+1}|X_k)$, also called proposal distribution, defined on the state space $\Omega$ and let $\rho_{BG}(x) = \exp(-U(x))/Z$ the BG ensemble we would like to sample from. Starting at $n = 0$, we define a Markov chain $X_k$ via the following repeated steps:*

1. *Given $X_k$, generate a proposal $X_{k+1}^{(p)}$ using the time evolution prescribed by $T$.*

2. *Compute the acceptance ratio*

$$A(X_{k+1}^{(p)}, X_k) = \arg\min\left\{1, \frac{\rho_{BG}(X_{k+1}^{(p)})g(X_k|X_{k+1}^{(p)})}{\rho_{BG}(X_k)g(X_{k+1}^{(p)}|X_k)}\right\} \tag{60}$$

3. *Sample a real number $u \sim \mathcal{U}[0,1]$. If $u < A(X_k, X_{k+1}^{(p)})$, accept the proposal and set $X_{k+1} = X_{k+1}^{(p)}$; otherwise, refuse the move and set $X_{k+1} = X_k$. Then, increment $n$ to $n + 1$.*

**Proposition 4.1.** *The Markov chain $X_k$ defined via MH algorithm has $\rho_{BG}(x)$ as unique stationary distribution, i.e.*

$$\rho_{BG}(x) = \int_{\Omega^d} T_{MH}(x|x')\rho_{BG}(x')dx', \qquad \forall x, x' \in \Omega \tag{61}$$

*where $T_{MH}(x|x')$ is the transition kernel of MH algorithm.*

*Proof.* We have show that (1) $\rho_{BG}(x)$ is a stationary distribution and (2) it is unique. Regarding (2) we advocate to geometric ergodicity(Mengersen & Tweedie, 1996). We present the proof of property (1): firstly, it is equivalent to detailed balance condition(Robert et al., 1999)

$$\rho_{BG}(x)T_{MH}(x, x') = \rho_{BG}(x')T_{MH}(x', x) \qquad \forall x, x' \in \Omega \tag{62}$$

The transition kernel is by definition

$$T_{MH}(x, x') = g(x'|x)A(x', x) + \delta(x - x')\left(1 - \int_\Omega A(x, s)g(s|x)ds\right) \tag{63}$$

where the first addend takes into account the case of accepted move, while the second of the rejected one. Actually, for $x = x'$ the detailed balance condition is trivially true. Then, for $x \neq x'$ we compute the left hand side of equation 62

$$
\begin{aligned}
\rho_{BG}(x)T_{MH}(x, x') &= \rho_{BG}(x)g(x'|x)A(x', x) \\
&= \rho_{BG}(x)g(x'|x)\arg\min\left\{1, \frac{\rho_{BG}(x')g(x|x')}{\rho_{BG}(x)g(x'|x)}\right\} \\
&= \arg\min\left\{\rho_{BG}(x)g(x'|x), \rho_{BG}(x')g(x|x')\right\}
\end{aligned}
\tag{64}
$$

The right hand side is symmetric with respect to swap of $x$ with $x'$, hence concluding the proof. $\qquad\square$

In practice, convergence is considered achieved when the acceptance ratio is consistently close to 1. In such cases, every newly generated proposal can be regarded as an independent sample obtained from $\rho_{BG}$.

Despite its popularity, the Metropolis-Hastings algorithm has some limitations. It is sensitive to the choice of the proposal distribution $g$ and its parameters, and improper tuning may result in inefficient exploration. For instance, in the so-called *random walk setting*, $g$ is chosen to be a Gaussian transition kernel, and its variance is a critical hyperparameter in this case. Moreover, the algorithm generates correlated samples, impacting the independence of successive samples and hindering accurate estimation even after convergence. Convergence may be slow in high-dimensional spaces, requiring numerous iterations. In such setups, the algorithm's performance is influenced by the initial state, and initial points far from the basin of the target may impede efficient exploration, leading to an acceptance rate close to zero. Another issue pertains to multimodal distributions, especially those with widely separated modes. They pose a significant challenge for the Metropolis-Hastings algorithm because, depending on the choice of $g$, jumps between modes can be very rare and may necessitate a very long chain to practically observe convergence.

The second class of Markov chain we would like to review are the Langevin-based algorithms. The basic idea is very close to the definition of naive score-based diffusion in Proposition 3.3. For the sake of simplicity let us fix the state space $\Omega = \mathbb{R}^d$.

**Proposition 4.2.** *Let us denote with $dW_t$ a Wiener process. Under Assumption 2.1, namely*

$$\exists a \in \mathbb{R}_+ \text{ and a compact set } \mathcal{C} \in \mathbb{R}^d : \ x \cdot \nabla U(x) \geq a|x|^2 \quad \forall x \in \mathbb{R}^d \setminus \mathcal{C}, \tag{65}$$

*the Langevin SDE*

$$dX_t = -\nabla U(x)dt + \sqrt{2}dW_t \qquad X_0 \sim \rho_0 \tag{66}$$

*have a global solution in law and is ergodic (Oksendal, 2003; Mattingly et al., 2002; Talay & Tubaro, 1990). For any initial condition $\rho_0(x)$ such solution is $\rho_{BG}(x)$.*

Given this result, one can define a Markov chain based on the time discretization of such SDE and use it for sampling(Parisi, 1981). Such procedure is commonly named Unadjusted Langevin Algorithm (ULA)(Roberts & Tweedie, 1996).

**Definition 4.3** (ULA). *Given a time step $h > 0$ and a set of i.i.d. gaussian variables $\{\xi_k\} \sim \mathcal{N}(\mathbf{0}_d, \boldsymbol{I}_d)$, the Unadjusted Langevin Algorithm (ULA) is the Markov chain defined as*

$$X_{k+1} = X_k - h\nabla U_{\theta_k}(X_k) + \sqrt{2h}\,\xi_k, \qquad\qquad X_0 \sim \rho_{\theta_0}, \tag{67}$$

*for $k \in \mathbb{N}$.*

Under Assumption 2.1, the Unadjusted Langevin Algorithm (ULA) is ergodic and possesses a unique global solution. An advantage over the Metropolis-Hastings (MH) algorithm is that the chain is uniquely defined via $U(x)$, and no proposal distribution is necessary. However, it is well-known that ULA represents a biased implementation of Langevin dynamics(Wibisono, 2018). For a nonzero time step, the global solution is $\rho_{bias} \neq \rho_{BG}$. Let us illustrate this point with a simple example.

**Example 4.1.** *Let $U(x) = (x-\mu)^T\Sigma^{-1}(x-\mu)/2 + \log[\det(2\pi\Sigma)]/2$, that is BG ensemble is a gaussian with mean $\mu$ and covariance matrix $\Sigma$. The associated Langevin SDE is also known as Ornstein-Uhlenbeck (OU) process(Uhlenbeck & Ornstein, 1930), having a linear drift as peculiarity:*

$$dX_t = -\Sigma^{-1}(X_t - \mu)dt + \sqrt{2}dW_t. \tag{68}$$

*It is possible to write an explicit solution using Ito integral, namely*

$$X_t - \mu \sim e^{-t\Sigma^{-1}}(X_0 - \mu) + \Sigma^{\frac{1}{2}}\left(\boldsymbol{I}_d - e^{-2t\Sigma^{-1}}\right)^{\frac{1}{2}} Z \tag{69}$$

*for any $t \geq 0$ and where $Z \sim \mathcal{N}(\mathbf{0}_d, \boldsymbol{I}_d)$ indipendently from $X_0$. It means that the law of the process converges exponentially fast to $\mathcal{N}(\mu, \Sigma)$. The associated ULA is*

$$X_{k+1} - \mu = \left(\boldsymbol{I}_d - h\Sigma^{-1}\right)(X_k - \mu) + \sqrt{2h}\xi_k. \tag{70}$$

*and the corresponding solution in law is*

$$X_k - \mu \sim A_h^k (X_0 - \mu) + \sqrt{2h} \left( \boldsymbol{I}_d - A_h^2 \right)^{-\frac{1}{2}} \left( \boldsymbol{I}_d - A_h^{2k} \right)^{\frac{1}{2}} Z \tag{71}$$

*where $A_h = \boldsymbol{I}_d - h\Sigma^{-1}$. Naming $\lambda_{min}(\Sigma) > 0$ the minimum eigenvalue of the covariance matrix, for $0 < h < \lambda_{min}(\Sigma)$ we have $\lim_{k \to \infty} A_h^k = 0$. Thus, for $k \to \infty$*

$$X_k \xrightarrow{d} \mu + \sqrt{2h} \left( \boldsymbol{I}_d - A_h^2 \right)^{-\frac{1}{2}} Z \tag{72}$$

*This means that the limiting measure for ULA is not $\rho_{BG}$, but*

$$\rho_{bias}(x) = \mathcal{N} \left( \mu, \Sigma \left( \boldsymbol{I}_d - \frac{h}{2}\Sigma^{-1} \right)^{-1} \right)(x) \tag{73}$$

This example illustrates that the Unadjusted Langevin Algorithm (ULA) exhibits bias even for a very simple target density. This phenomenon has been recently analyzed mathematically(Wibisono, 2018). The physical interpretation is that detailed balance is broken by construction. Let us elaborate on this point: in Proposition 3.2, we demonstrated how a Stochastic Differential Equation (SDE) can be associated with a Partial Differential Equation (PDE). The specific case studied in this section was previously analyzed in Proposition 3.3. Specifically, the Boltzmann-Gibbs (BG) density is the unique minimizer of the Kullback-Leibler (KL) divergence functional $D_{\mathrm{KL}}(\rho \parallel \rho_{GB})$. Moreover, the Fokker-Plank PDE corresponds to the gradient flow in $\mathcal{P}(\mathbb{R}^d)$ with respect to the 2-Wasserstein distance $\mathcal{W}_2$(Jordan et al., 1998). If we split equation 67 in two substeps

$$\begin{aligned} X_{k+\frac{1}{2}} &= X_k - h\nabla U(X_k) \\ X_{k+1} &= X_{k+\frac{1}{2}} + \sqrt{2\varepsilon}\xi_k \end{aligned} \tag{74}$$

we can associate each of them to a precise operation in probability space. In particular, denoting with $\rho_i$ the law of $X_i$, we obtain

$$\begin{aligned} \rho_{k+\frac{1}{2}} &= (\boldsymbol{I}_d - h\nabla U)_{\#}\rho_k \\ \rho_{k+1} &= \mathcal{N}(\boldsymbol{0}_d, 2h\boldsymbol{I}_d) \star \rho_{k+\frac{1}{2}} \end{aligned} \tag{75}$$

We recall the decomposition of the Kullback-Leibler (KL) divergence as $D_{\mathrm{KL}}(\rho \parallel \rho_{GB}) = -H(\rho, \rho_{GB}) - H(\rho)$. In equation 75, the first step involves the forward discretization of gradient descent on $-H(\rho, \rho_{GB}) = \mathbb{E}_{\rho}[U]$, while the second step represents the exact gradient flow for negative entropy in probability space. Therefore, ULA is also referred to as the Forward-Flow method in probability space. The bias arises because the forward gradient descent does not correspond, in probability space, to the adjoint of the flow at iteration $k+1/2$. One possible solution is to use forward-backward combinations, referring to proximal algorithms(Parikh et al., 2014). In particular, the Forward-Backward (FB) implementation for Langevin dynamics would be

$$\begin{aligned} \rho_{k+\frac{1}{2}} &= (\boldsymbol{I}_d - h\nabla U)_{\#}\rho_k \\ \rho_{k+1} &= \arg\min_{\rho \in \mathcal{P}} \left\{ -H(\rho) + \frac{1}{2\epsilon}\mathcal{W}_2\left(\rho, \rho_{k+\frac{1}{2}}\right)^2 \right\} \end{aligned} \tag{76}$$

Similarly, the Backward-Forward (BF) version

$$\begin{aligned} \rho_{k+\frac{1}{2}} &= \left((\boldsymbol{I}_d + h\nabla U)^{-1}\right)_{\#}\rho_k \\ \rho_{k+1} &= \exp_{\rho_{k+\frac{1}{2}}}\left(-h\nabla \log \rho_{k+\frac{1}{2}}\right) \end{aligned} \tag{77}$$

where exp is the exponential map. Unfortunately, both FB and BF are not implementable in practice, except for the trivial case of gaussian initial data and target $\rho_{BG}$. The heat flow (the step $k + 1/2$) is the most problematic since it concerns steps in probability space. Neither forward (n.b. beyond one iteration) nor

backward are usable. As a side note, one could imagine to directly perform a single forward or backward step on the KL divergence. Unfortunately, the encountered issues are the same one has for the heat flow, i.e. the hard task appears to be the actual implementation of forward or backward routines in probability space. In conclusion, ULA appears to be the simplest time discretization of Langevin dynamics, since it is practically implementable in general, hence very used for sampling from a BG ensemble. However, it is known to be biased and other methods are studied to eliminate, or at least reduce, such bias.

One possibility we would like to review is Metropolis Adjusted Langevin Algorithm(Grenander & Miller, 1994) (MALA), which represents a sort of hybrid between MH and ULA.

**Definition 4.4** (MALA). *Metropolis Adjusted Langevin Algorithm is a particular case of MH algorithm 4.2 where the proposal distribution is the transition kernel associated to ULA equation 67, namely (for $x \in \mathbb{R}^d$)*

$$g(x' \mid x) = \frac{1}{(2\pi h)^{\frac{d}{2}}} \exp\left(-\frac{1}{4h}\|x' - x + hU(x)\|_2^2\right) \tag{78}$$

On the other hand, one can interpret MALA as a corrective measure for the breakdown of detailed balance in ULA. While the Metropolis-Hastings algorithm inherently respects detailed balance, implying that MALA becomes asymptotically unbiased for a large number of iterations as $k \to \infty$, certain challenges persist. A primary concern is the sensitivity to the choice of the step size $h$ during the discretization of Langevin dynamics, significantly influencing the efficiency of sampling. When $h$ is too small, it can lead to poor exploration and potentially a very low acceptance rate, while an excessively large $h$ can lead to instability of the chain. Determining an optimal $h$ lacks a general rule, contributing to MALA introducing bias in samples, particularly evident when the target distribution features sharp peaks or multimodal structures. This bias introduces potential inaccuracies in statistical estimates.

In practical applications, MALA may exhibit random walk behavior, especially when step sizes are inadequately tuned, resulting in inefficient exploration and sluggish convergence. The algorithm's performance is further contingent on the choice of initial conditions, and beginning far from high-probability regions may necessitate a considerable number of iterations for meaningful exploration. Additionally, MALA may struggle to adapt to changes in the geometry of the target distribution, particularly when facing varying curvatures or strong anisotropy.

While various more advanced algorithms exist, they often build upon the foundational concepts discussed in this section. Notable among them is Hamiltonian (or Hybrid) Monte Carlo(Duane et al., 1987) (HMC), an advanced MCMC method inspired by Hamiltonian mechanics. HMC utilizes fictitious Hamiltonian dynamics to propose new states, enhancing exploration, especially in high-dimensional systems. Gibbs sampling(Geman & Geman, 1984), another MCMC approach, iteratively samples from conditional distributions given current variable values on a single dimension, proving effective, particularly in high-dimensional spaces. Parallel Tempering, or Replica Exchange(Swendsen & Wang, 1986), involves running multiple chains at different temperatures concurrently, with periodic swaps between neighboring chains to facilitate improved exploration.

In general, most methods aim to find a chain that produces independent samples from a Boltzmann-Gibbs ensemble, particularly when run for extended periods. A critical issue is measuring the effective bias due to the truncation at finite time of the chain, posing challenges for convergence towards the asymptotic $\rho_{BG}$. Unfortunately, few general results are available, and they are often limited to specific BG ensembles, such as Gaussian or log-concave densities. This becomes particularly problematic in the context of Energy-Based Models (EBM), as outlined in Remark 2.2, where sampling from a BG distribution is required at each step of parameter optimization.

## 5 EBMs and physics

In this section, we review the Boltzmann-Gibbs ensemble in the context of equilibrium Statistical Physics, deriving it through the maximum entropy principle. This approach provides a mathematically precise yet concise justification for the Boltzmann distribution, which underpins the probabilistic structure of EBMs. Our goal is to offer a compact introduction to key statistical physics concepts, such as free energy and partition functions, that recur in the discussion of EBMs. While this derivation is not directly related to the

practical training of an EBM, we believe it is important to equip the reader of this review with the necessary physical and mathematical tools to understand recurring concepts in EBMs, such as the Boltzmann-Gibbs distribution, entropy, and free energy.

The first step involves the derivation of the Boltzmann-Gibbs ensemble. Here, we present a derivation based on information theory(Jaynes, 1957a;b), offering a posteriori physical interpretation of the quantities we will manipulate. Alternative methods of proof are also available(Gallavotti, 1999). Consider a physical system whose state is uniquely determined by a variable $x \in \Omega \subseteq \mathbb{R}^d$, where $\Omega$ is often referred to as phase space. The connection with information theory is linked to the fundamental problem of Statistical Physics: describing a system as a statistical ensemble, i.e., identifying an observation of $x$ as a sample from an underlying PDF $\rho$. Like classical statistics, $\rho$ contains a wealth of information about the system, particularly its global properties.

In Physics, this dichotomy translates into the microscopic versus macroscopic realms. Let's envision a simple thought experiment: picture a large city where each of the $N$ inhabitants is given a fair coin, with the coin's state represented by our variable $x \in \{-1, 1\}^N$. Twice a day, everyone has to flip their coin. If we were omniscient, there would be a way to predict the state $x$ with no error (i.e., the microscopic state) and derive any global (macroscopic) property, such as the sum or product of the state values at each flipping event, with no error. However, in reality, nobody could achieve this; we rely on statistics, the central limit theorem, and so forth. In other words, we know the probability density $\rho(x)$ from which the process is a sampled event. For instance, if $N$ is large enough, we expect the average of the state vector to be 0 for any flipping event, and we can deduce so directly from $\rho$.

In Statistical Physics, each coin represents a component of a system, such as a particle in a gas, for which a direct measurement of $x$ is unattainable. The goal is to determine $\rho$ so that standard statistical tools can be used to analyze global properties. The challenge that makes Statistical Physics more complex than the simple example above is that the dynamics of individual components can be unknown and inaccessible. Additionally, interactions between components can make the identification of $\rho$ challenging, even if the underlying microscopic dynamics are known.

To address this issue, we recognize that, before formulating any physical model, we need some motivated assumptions—constraints or information—regarding how the system should behave, at least on a macroscopic level. This is the bare minimum; without any information about a system, it is impossible to provide any meaningful analysis. Thus, adopting a claim of epistemic modesty, one can state that we aim to select the model compatible with such constraints that maximizes our ignorance about the system. The mathematical translation of such idea is the *Principle of Maximum Entropy.*

**Assumption 5.1** (Principle of Maximum Entropy). *Let us consider the unknown $\rho : \Omega \to \mathbb{R}_+$ that describes the probability distribution of the states. We assume $\rho$ to be absolutely continuous w.r.t. Lebesgue measure without loss of generality. Given a vector field $F : \Omega \to \mathbb{R}^d$, $\Lambda \in \mathbb{R}^d$ and a PDF $\pi$, a set of constraints is any component-wise (in)equality*

$$I_k[\pi] := \int_\Omega F_k(x)\pi(x)dx \leq_k \Lambda_k \qquad k = 1, \ldots, d \tag{79}$$

*where the symbol $\leq_k$ can be an equality or inequality. The* **Principle of Maximum Entropy** *is*

$$\rho = \underset{\substack{\pi \in \mathcal{P}(\Omega) \\ I_k[\pi] \leq_k \Lambda_k}}{\arg\max} H(\pi) \tag{80}$$

*where $H(\rho)$ is the usual differential entropy, cfr. equation 7.*

This variational formulation identifies the "ignorance" about the system with the entropy associated to $\rho$. It has been proven that the entropy can be characterized in an axiomatic way(Aczél et al., 1974), so that the definition of differential entropy is unique with respect to certain properties. For the sake of the present treatment, let us motivate the Maximum Principle with a simple example.

**Example 5.1** (Maximum principle on an interval). *Let us consider an interval $\Omega = [a, b] \subset \mathbb{R}$, with $\text{Vol}(\Omega) = \int_a^b dx$. Moreover, the only constraint is that $\rho$ can be normalized and is positive. Thus, we*

*have $I_0[\pi] = \int_a^b \rho(x)dx = 1$ and*

$$\rho = \underset{\substack{\pi \in \mathcal{P}(\Omega) \\ I_0[\pi]=1, \ \pi>0}}{\arg\max} \ H(\pi) \tag{81}$$

*We can use Lagrange multipliers to solve a constrained optimization problem, solving the unconstrained optimization of the Lagrangian*

$$J(\pi) := H(\pi) + \lambda_0 \left( \int_a^b \pi(x)dx - 1 \right) \tag{82}$$

*To find stationary points we can compute the first variational derivative with respect to $\pi$ and finding its roots, namely solutions of*

$$\frac{\delta J(\pi)}{\delta \pi} = -\log \pi - 1 + \lambda_0 = 0 \tag{83}$$

*that is $\rho(x, \lambda_0) = \exp(\lambda_0 - 1)$. To find $\lambda_0$ we can substitute $\rho(x, \lambda_0)$ into the constraint, yielding $\lambda_0 = 1 - \log(b - a)$. In conclusion, $\rho(x) = 1/(b - a)$, which also satisfies the positivity request. We have just to check that such stationary point is a maximum. The second variation of $J(\pi)$ evaluated in the stationary point is*

$$\left.\frac{\delta^2 J(\pi)}{\delta \pi^2}\right|_{\pi=\rho} = -\frac{1}{\rho(x)} < 0 \tag{84}$$

*Hence, we conclude that $\rho(x)$ is a maximum. If $\Omega$ is discrete such derivation can be easily generalized. The interpretation is straightforward: imagine that $\Omega$ is the event space for some random process. Without any knowledge, the simplest possible model is the one that associates the same probability to all the possible outcomes.*

At this point we have all the ingredients to present the derivation of Boltzmann-Gibbs ensemble.

**Proposition 5.1** (Boltzmann-Gibbs ensemble from Maximum Entropy principle). *Given $U(x)$ that satisfies Assumption 2.1 and a constant $U^*$, the Boltzmann-Gibbs $\rho_{BG} = e^{\lambda_1 U(x)}/Z$, where $\lambda_1 > 0$, is the unique solution of the variational maximization problem equation 80 where the set of constraints is*

$$\begin{aligned} I_0[\pi] &:= \int_\Omega \pi(x)dx = 1 \\ I_1[\pi] &:= \int_\Omega U(x)\pi(x)dx = U^* \end{aligned} \tag{85}$$

*Proof.* The proof proceeds similarly to Example 5.1. The constrained optimization problem equation 80 is associated to the unconstrained one

$$\rho = \underset{\pi \in \mathcal{P}(\Omega)}{\arg\max} J(\pi) := \underset{\pi \in \mathcal{P}(\Omega)}{\arg\max} H(\pi) + \lambda_0 \left( \int_\Omega \pi(x)dx - 1 \right) + \lambda_1 \left( \int_\Omega U(x)\pi(x)dx - U^* \right). \tag{86}$$

where $\lambda_0, \lambda_1$ are Lagrange multiplier. We compute the first variational derivative and find its roots

$$\frac{\delta J(\pi)}{\delta \pi} = -\log \pi(x) - 1 + \lambda_0 + \lambda_1 U(x) = 0 \tag{87}$$

That is, $\rho(x) = \exp(\lambda_0 + \lambda_1 U(x) - 1)$. This means that $\lambda_0$ can be incorporated in the normalization factor, namely the partition function $Z^{-1} = \exp(\lambda_0 - 1)$, and determined via the constraint $I_0[\rho] = 1$. While $\lambda_1$ is implicitly determined by $I_1$. To check that such solution is a maximum, we compute the second variation obtaining a result analogous to equation 84. $\square$

The remaining issue is the identification of $\lambda_1$ with $\beta = 1/k_B T$, related to temperature $T$ and Boltzmann dimensional constant $k_B = 1.23 \times 10^{-28} \, \mathrm{J} \cdot \mathrm{K}^{-1}$. The reason is that if we interpret the Boltzmann-Gibbs ensemble with an equilibrium ensemble, the derivation via Maximum Entropy principle must be coherent with Thermodynamics(Adkins, 1983). A complete survey on such field would lead the present treatment

off-topic. The take home message is related to a different interpretation of the unconstrained optimization problem equation 86, namely

$$\frac{\delta}{\delta\pi}\left(H(\pi) + \lambda_0 \int_\Omega \pi(x)dx + \lambda_1 \int_\Omega U(x)\pi(x)dx\right) = 0 \tag{88}$$

In particular, the following lemma holds true:

**Lemma 5.1.** *Maximum Entropy principle and its variational formulation are equivalent to*

- *Minimization of energy functional $\bar{U} = \int_\Omega U(x)\pi(x)dx$ under the constraints*

$$I_0[\pi] \coloneqq \int_\Omega \pi(x)dx = 1$$
$$I_1[\pi] \coloneqq H(\pi) = H^* > 0 \tag{89}$$

- *Minimization of Helmholtz Free Energy functional $F = \bar{U} - TH(\pi)$, where $T > 0$ is the usual thermodynamic temperature, under the constraint*

$$I_0[\pi] \coloneqq \int_\Omega \pi(x)dx = 1 \tag{90}$$

*Proof.* The proof is just related to a redefinition of the Lagrange multipliers, leading to the same eq. 87 for the stationarity condition. For the minimization of energy, one define $\lambda_1' = -1/\lambda_1$ and $\lambda_0' = -\lambda_0/\lambda_1$, where the sign is just a convention, obtaining

$$\frac{\delta}{\delta\pi}\left(\lambda_1' H(\pi) + \lambda_0' \int_\Omega \pi(x)dx + \int_\Omega U(x)\pi(x)dx\right) = 0 \tag{91}$$

While for the Helmholtz Free Energy, we have just to impose the thermodynamics constraint(Fermi, 2012) $\partial F/\partial S = -T$, that is $\lambda_1 = -1/T$. □

**Remark 5.1** (Free Energy in EBM training)**.** *In Section 2.1 we presented the training procedure for an EBM as the KL divergence minimization. If $\rho_\theta$ is in the same class of $\rho_*$, the global minimum in probability space corresponds to $\rho_\theta = \rho_*$, i.e. KL divergence equal to zero since by definition $D_{KL}(\rho_\theta \| \rho_\theta) = 0$. However, if we expand the such identity, we have*

$$\log Z_\theta + \beta \int_{\mathbb{R}^d} U_\theta(x)\rho_\theta(x)dx - H(\rho_\theta) = 0 \tag{92}$$

*where we used equation 10, and restored $\beta$ in front of $U_\theta$ (n.b. we put $k_B = 1$ and $T = 1$ in Section 2.1). If we identify $F_\theta = -\log Z_\theta$, we immediately notice that equation 92 is the definition of Helmholtz Free Energy. In fact, the convergence of the training corresponds to have reached the equilibrium. The equality $F = \bar{U} - TH(\pi)$ is not true out of equilibrium — the KL divergence $D_{KL}(\rho_\theta \| \rho_*)$ is zero iff $\rho_\theta = \rho_*$. Moreover, it is even more clear the statement of Lemma 5.1: since*

$$\log Z_\theta + \min_{\substack{\pi \in \mathcal{P}(\Omega) \\ I_0[\pi]=1,\, \rho_\theta > 0}} [\beta \int_{\mathbb{R}^d} U_\theta(x)\pi(x)dx - H(\pi)] = 0 \tag{93}$$

*and $F = \bar{U} - TH(\pi)$, at equilibrium $F$ is necessarily minimized in correspondence of $F[\rho_\theta] = -\log Z_\theta$.*

The requested compatibility between Thermodynamics and the Maximum Entropy principle in Lemma 5.1 represents the final ingredient needed to define the Boltzmann-Gibbs probability density associated with a system at equilibrium with a thermal reservoir at temperature $T$. For the purpose of this review, it would be beneficial to elaborate on the physical significance of EBM. Assuming we are dealing with an equilibrium ensemble, we presume that the parameters $\theta$ in the energy $U_\theta$ have already been determined. Similar to a physical gas where particles move within an energy landscape, different datasets or even individual

data points can be envisioned as snapshots of an evolving physical system. The crucial aspect is that from a statistical perspective, the average energy $\bar{U}$ associated with the EBM must remain constant, with fluctuations suppressed as the number of components increases. An example of dynamics consistent with such a constraint is Langevin dynamics. The connection with sampling and Physics becomes evident: sampling is the process of relaxation(Evans et al., 2009) towards equilibrium. Utilizing our understanding of nature entails designing sampling routines capable of facilitating such relaxation.

We introduced the concept of free energy as a thermodynamic quantity minimized at equilibrium by the Boltzmann-Gibbs ensemble. Generally, computing free energy is a extensively studied problem in Chemistry(Jorgensen, 1989), spanning from organic Chemistry to protein folding(Dinner et al., 2000). However, the concept of free energy appears ubiquitous, extending into seemingly disparate contexts far from computational chemistry, such as autoencoders(Hinton & Zemel, 1993), lattice field theory(Nicoli et al., 2021), and neuroscience(Friston, 2009). Invariably, it is associated with some equilibrium principle, often directly linked to the use of a generalization of the Boltzmann-Gibbs ensemble.

The importance of free energy can be readily understood: the expected value of any observable at equilibrium can be computed if we have access to the normalization constant of the Boltzmann-Gibbs ensemble, which is the partition function $Z = e^{-F}$. However, as demonstrated in Section 2.1, computing the partition function, and consequently the free energy, is exceedingly complex using standard Monte Carlo methods for systems with many degrees of freedom, roughly corresponding to dimension $d$ for EBM training. Among the various proposed advanced methods(Stoltz et al., 2010), the utilization of the Jarzynski identity(Jarzynski, 1997) stands out a very notable tool. Recently, an application of such result for improving the training of EBMs has been proposed, cfr. (Carbone et al., 2024a) and next Section.

## 6 Recent Developments in EBM training

In this section we summarize the most common algorithms used for EBM training and some recent developments in this area. In the first part, we will discuss the most common approach to directly minimize the KL divergence loss, which is Contrastive Divergence. In the second part, we will briefly present the so-called sampling-free methods, namely Score-Matching and Noise Constrastive Estimation.

### 6.1 Sampling-based methods

For readers convenience, we fix the notation: in the following, $\rho_\theta(x) = \rho_{\theta(t)}(x) = \exp(-U_{\theta(t)}(x))/Z_\theta$ is the EBM we aim to train. As we showed in Section 2, training an EBM reduces to perform gradient-based optimization on cross-entropy. After some manipulation, the gradient of $H(\rho_*, \rho_\theta)$ reduces to

$$\partial_\theta H(\rho_*, \rho_\theta) = \mathbb{E}_*[\partial_\theta U_\theta] - \mathbb{E}_\theta[\partial_\theta U_\theta] \coloneqq -\mathcal{D}. \tag{94}$$

As we stressed in Remark 2.2, the main issue is the estimation of $\mathbb{E}_\theta[\partial_\theta U_\theta]$. An analytical computation is outreach for a generic $U_\theta$, as well as the use of numerical spline methods which are impractical in high dimension. The only possibility is to generate a set of $N$ samples $\{X^i\}_{i=1}^N$ distributed as $\rho_{\theta(t)}$ and exploit a Monte Carlo integration, namely

$$\mathbb{E}_\theta[\partial_\theta U_\theta] \approx \frac{1}{N} \sum_{i=1}^N \partial_\theta U_\theta(X^i) \qquad X^i \sim \rho_\theta \tag{95}$$

We stress that such generation is required at *each optimization step* of $\theta$. The basic idea is to couple a gradient-based routine with a Markov Chain (Liu, 2001) devoted to the generation of the needed samples (see Section 4 for a detailed description of sampling). Without loss of generality, we present the state-of-the-art algorithm using Unadjusted Langevin algorithm (ULA) (Parisi, 1981) as the sampler.

As mentioned, a problem encountered by standard sampling routines (such as ULA) is related to multimodality; that is, for fixed $\theta$ and a general initial condition $\bar{X} \sim \bar{\pi}$ for the Markov Chain, there are no general results on the convergence rate towards the desired equilibrium $X \sim \rho_\theta$. However, if one were to

choose a smart initial condition, such an issue is alleviated. For instance, in the ideal case where we could sample from an initial distribution $\bar{\rho}$ very close to $\rho_\theta$. In this sense, the naive approach in which the sampling routine restarts from the same "simple" distribution, like a Gaussian, for every optimization step, is not well adapted to EBM training. The question then arises: how to select an appropriate initial condition?

The idea of Contrastive Divergence(Hinton, 2002) (CD) and Persistent Contrastive Divergence(Tieleman, 2008) (PCD) in their original formulation is to use the unknown data distribution $\rho_*$ as the initial condition for Markov Chain sampler. This is feasible since we have the dataset; that is, we could simply extract some data points from it and use them as the initial condition of the sampler at every optimization step. To better analyze the two routines, we present CD and PCD in Algorithms 1 and 2, where ULA is chosen as the sampling routine.

---

**Algorithm 1** Contrastive divergence (CD) algorithm

---

1: **Inputs:** data points $\mathcal{X} = \{x_*^i\}_{i=1}^n$ in $\mathbb{R}^d$; energy model $U_\theta$; optimizer step $\mathrm{opt}(\theta, \mathcal{D})$ using $\theta$ and the empirical gradient $\mathcal{D}$; initial parameters $\theta_0$; number of walkers $N \in \mathbb{N}_0$ with $N < n$; total duration $K \in \mathbb{N}$; ULA time step $h$; $P \in \mathbb{N}$.
2: **for** $k = 1, \ldots, K-1$ **do**
3:     **for** $i = 1, ..., N$ **do**
4:         $X_0^i = RandomSample(\mathcal{X})$
5:         **for** $p = 0, ..., P-1$ **do**
6:             $X_{p+1}^i = X_p^i - h\nabla U_{\theta_k}(X_p^i) + \sqrt{2h}\,\xi_p^i, \qquad \xi_p^i \sim \mathcal{N}(0_d, I_d)$ $\qquad\qquad$ ▷ ULA
7:         **end for**
8:     **end for**
9:     $\tilde{\mathcal{D}}_k = N^{-1}\sum_{i=1}^N \partial_\theta U_{\theta_k}(X_P^i) - n^{-1}\sum_{i=1}^n \partial_\theta U_{\theta_k}(x_*^i)$ $\qquad\qquad$ ▷ empirical gradient
10:     $\theta_{k+1} = \mathrm{opt}(\theta_k, \mathcal{D}_k)$ $\qquad\qquad$ ▷ optimization step
11: **end for**
12: **Outputs:** Optimized energy $U_{\theta_K}$; set of walkers $\{X_P^i\}_{i=1}^N$

---

**Algorithm 2** Persistent contrastive divergence (PCD) algorithm

---

1: **Inputs:** data points $\mathcal{X} = \{x_i^*\}_{i=1}^n$ in $\mathbb{R}^d$; energy model $U_\theta$; optimizer step $\mathrm{opt}(\theta, \mathcal{D})$ using $\theta$ and the empirical CE gradient $\mathcal{D}$; initial parameters $\theta_0$; number of walkers $N \in \mathbb{N}_0$ with $N < n$; total duration $K \in \mathbb{N}$; ULA time step $h$.
2: $X_0^i = RandomSample(\mathcal{X})$ for $i = 1, \ldots, N$.
3: **for** $k = 1, \ldots, K-1$ **do**
4:     $\tilde{\mathcal{D}}_k = N^{-1}\sum_{i=1}^N \partial_\theta U_{\theta_k}(X_k^i) - n^{-1}\sum_{i=1}^n \partial_\theta U_{\theta_k}(x_*^i)$ $\qquad\qquad$ ▷ empirical gradient
5:     $\theta_{k+1} = \mathrm{opt}(\theta_k, \mathcal{D}_k)$ $\qquad\qquad$ ▷ optimization step
6:     **for** $i = 1, ..., N$ **do**
7:         $X_{k+1}^i = X_k^i - h\nabla U_{\theta_k}(X_k^i) + \sqrt{2h}\,\xi_k^i, \qquad \xi_k^i \sim \mathcal{N}(0_d, I_d)$ $\qquad\qquad$ ▷ ULA
8:     **end for**
9: **end for**
10: **Outputs:** Optimized energy $U_{\theta_K}$; set of walkers $\{X_K^i\}_{i=1}^N$.

---

Let us clarify the notation. Each $X$ used for the estimation of the gradient of cross-entropy is named a "walker". Each walker is indexed by a superscript, and the function $RandomSample(\mathcal{X})$ performs a random extraction of $N$ points from $\mathcal{X}$. In CD, the chain for sampling is reinitialized at data at every cycle; in PCD, as for the name, the chain is "persistent", meaning it starts from the data just at the first iteration — after each optimization step, the sampling routine restarts from the samples found at the previous iteration. Traditionally, $x^*$ is referred to as "positive" samples, while the samples from $\rho_\theta$ are termed "negative", especially in the community of Boltzmann Machines. The adjective "Contrastive" originates from the minus sign between expectations in equation 94: the contribution of negative and positive samples to the variation of cross-entropy is indeed opposite. In fact, the ODE associated to gradient descent on cross-entropy minimization is

$$\dot{\theta} = \mathbb{E}_\theta[\partial_\theta U_\theta] - \mathbb{E}_*[\partial_\theta U_\theta] \tag{96}$$

This equation can be interpreted as gradient descent on the energy per positive sample and gradient ascent for the energy per negative sample. It corresponds to increasing the probability of data points in the dataset and decreasing it for the samples obtained from the chain. Stationarity is reached when $\rho_* = \rho_\theta$, so that generated points belong to the same distribution as true data points.

The natural question that arises concerns the convergence of the algorithms. To simplify the treatment, we do not analyze the algorithms for a finite set of walkers, but we study the time evolution of the probability distribution of the walkers $\check{\rho}(t, x)$. Ideally, this should remove any possible spurious bias from the analysis and permit an easier analytical study. We can write down an equation that mimics the evolution of the PDF of the walkers in the CD algorithm in the continuous-time limit. This equation reads:

$$\partial_t \check{\rho} = \alpha \nabla \cdot \left(\nabla U_{\theta(t)}(x)\check{\rho} + \nabla\check{\rho}\right) - \nu(\check{\rho} - \rho_*), \qquad \check{\rho}(t = 0) = \rho_* \tag{97}$$

with fixed $\alpha > 0$ and where the parameter $\nu > 0$ controls the rate at which the walkers are reinitialized at the data points: the last term in equation 97 is a birth-death term that captures the effect of these reinitializations. The solution to this equation is not available in closed from (and $\check{\rho}(t, x) \neq \rho_{\theta(t)}(x)$ in general), but in the limit of large $\nu$ (i.e. with very frequent reinitializations), we can show(Domingo-Enrich et al., 2021) that

$$\check{\rho}(t, x) = \rho_*(x) + \nu^{-1}\alpha\nabla \cdot \left(\nabla U_{\theta(t)}(x)\rho_*(x) + \nabla\rho_*(x)\right) + O(\nu^{-2}). \tag{98}$$

As a result, the gradient of cross-entropy equation 94 is

$$\int_{\mathbb{R}^d} \partial_\theta U_{\theta(t)}(x)(\rho_*(x) - \check{\rho}(t, x))dx$$
$$= -\nu^{-1} \int_{\mathbb{R}^d} \partial_\theta U_{\theta(t)}(x)\nabla \cdot \left(U_{\theta(t)}(x)\rho_*(x) + \nabla\rho_*(x)\right)dx + O(\nu^{-2}) \tag{99}$$
$$= \nu^{-1} \int_{\mathbb{R}^d} \left(\partial_\theta\nabla U_{\theta(t)}(x) \cdot \nabla U_{\theta(t)}(x) - \partial_\theta\Delta U_{\theta(t)}(x)\right)\rho_*(x)dx + O(\nu^{-2})$$

The leading order term at the right hand side is precisely $\nu^{-1}$ times the gradient with respect to $\theta$ of the Fisher divergence

$$\frac{1}{2}\int_{\mathbb{R}^d} |\nabla U_\theta(x) + \nabla\log\rho_*(x)|^2\rho_*(x)dx$$
$$= \frac{1}{2}\int_{\mathbb{R}^d} \left[|\nabla U_\theta(x)|^2 - 2\Delta U_\theta(x) + |\nabla\log\rho_*(x)|^2\right]\rho_*(x)dx \tag{100}$$

where $\Delta$ denotes the Laplacian and we used

$$\int_{\mathbb{R}^d} \nabla U_\theta(x) \cdot \nabla\log\rho_*(x)\rho_*(x)dx = \int_{\mathbb{R}^d} \nabla U_\theta(x) \cdot \nabla\rho_*(x)dx = -\int_{\mathbb{R}^d} \Delta U_\theta(x)\rho_*(x)dx \tag{101}$$

This confirms the known fact that the CD algorithm effectively performs GD on the Fisher divergence rather than the cross-entropy(Hyvarinen, 2007), similarly to score matching.

Regarding PCD, the associated PDE is equation 97 with $\nu = 0$. Again, the solution $\check{\rho}(t, x) \neq \rho_{\theta(t)}(x)$ in general, thus for any finite $\alpha$, we have $\mathbb{E}_{\check{\rho}}[\partial_\theta U_\theta] \neq \mathbb{E}_\theta[\partial_\theta U_\theta]$. In other words, one cannot be sure to perform true gradient descent on cross-entropy — if we were able to estimate the loss, we could observe non-monotonic behavior. Extensions of standard PCD exploit an initial condition different from $\rho_*$ for the persistent chain, but such approach is plagued by the same issue regarding the convergence rate towards equilibrium.

Neither Contrastive Divergence nor Persistent Contrastive Divergence perform true gradient-based optimization of cross-entropy. The bias in both methods is inherently tied to time scales: in PCD, to the length of the Markov chain used for sampling, and in CD, to the reinitialization frequency. Despite their widespread use, this fundamental limitation affects the applicability of Energy-Based Models, making their performance highly problem-dependent. A key takeaway is that while generating individual samples is not an issue, CD and PCD may fail to capture the global mass distribution due to the properties of Fisher divergence, particularly in multimodal distributions. Moreover, both methods do not provide a "full" EBM since they do not provide any estimate of the partition function, critically affecting the advantages of EBMs that have been discussed in previous sections.

Alternative training methods leveraging nonequilibrium statistical physics have been recently explored in (Carbone et al., 2024a;b), offering a detailed comparison with CD/PCD and extensive numerical experiments. In particular, these approaches employ fluctuation theorems, such as the discrete-time Jarzynski equality, to correct for the bias introduced by short Markov chains and finite discretization steps in ULA.

The key idea is that the parameters $\theta$ are evolved along a protocol $\{\theta_k\}_{k \in \mathbb{N}_0}$, while the samples $X_k$ and associated nonequilibrium weights $A_k$ are iteratively generated by:

$$\begin{cases} X_{k+1} = X_k - h\nabla U_{\theta_k}(X_k) + \sqrt{2h}\,\xi_k, & X_0 \sim \rho_{\theta_0}, \\ A_{k+1} = A_k - \alpha_{k+1}(X_{k+1}, X_k) + \alpha_k(X_k, X_{k+1}), & A_0 = 0, \end{cases} \qquad (102)$$

where $\alpha_k(x, y)$ is defined by:

$$\alpha_k(x, y) = U_{\theta_k}(x) + \tfrac{1}{2}(y - x) \cdot \nabla U_{\theta_k}(x) + \tfrac{1}{4}h|\nabla U_{\theta_k}(x)|^2. \qquad (103)$$

Despite the fact that the sampling dynamics is based on the unadjusted Langevin algorithm (ULA), which introduces discretization error and mixing inefficiency, the inclusion of the weights $A_k$ allows for an *exact* expression of the model gradient:

$$\mathbb{E}_{\theta_k}[\partial_\theta U_{\theta_k}] = \frac{\mathbb{E}[\partial_\theta U_{\theta_k}(X_k)e^{A_k}]}{\mathbb{E}[e^{A_k}]}, \qquad Z_{\theta_k} = Z_{\theta_0}\mathbb{E}\left[e^{A_k}\right]. \qquad (104)$$

This reweighting strategy ensures that sampling errors are corrected and enables unbiased learning of EBMs via the update:

$$\theta_{k+1} = \theta_k + \gamma_k \mathcal{D}_k, \qquad \mathcal{D}_k = -\partial_\theta H(\rho_{\theta_k}, \rho_*) = \frac{\mathbb{E}[\partial_\theta U_{\theta_k}(X_k)e^{A_k}]}{\mathbb{E}[e^{A_k}]} - \mathbb{E}_*[\partial_\theta U_{\theta_k}], \qquad (105)$$

where $\gamma_k > 0$ is the learning rate. This update can also be implemented with adaptive optimizers such as AdaGrad or ADAM using the unbiased gradient $\mathcal{D}_k$. The interpretation is that reweighted samples are *always* in equilibrium with the time-evolving Boltzmann-Gibbs density during training, allowing to compute exactly the problematic expectation in the gradient of KL; the presence of the weights corrects the bias present in CD and PCD. More importantly, this framework allows to recursively tracking the cross-entropy via

$$H(\rho_{\theta_k}, \rho_*) = \log \mathbb{E}[e^{A_k}] + \log Z_{\theta_0} + \mathbb{E}_*[U_{\theta_k}], \qquad (106)$$

and the normalization constant $Z_k$. The former is a principled, tractable metric of training progress, while the latter is important for interpretability features of the trained EBM, as previously discussed. These developments open new directions for scalable, statistically consistent training of EBMs, with applications in fields where likelihood-based evaluation and calibrated uncertainty quantification are essential. Similar ideas were developed later in the context of sampling, where the target density is known analytically, but no data samples are available, cfr. Albergo & Vanden-Eijnden (2024); Vargas et al. (2024).

## 6.2 Sampling-free methods

From the previous treatmentm, it is evident that sampling-based methods such as CD or PCD rely on approximate sampling from the model distribution $\rho_\theta(x) \propto e^{-U_\theta(x)}$ and introduce several issues: (i) they require careful tuning of hyperparameters like the step size and number of iterations; (ii) the resulting gradients can be biased due to finite-length chains; and (iii) the computational cost can be prohibitive in high-dimensional or multi-modal settings.

In response to these challenges, a class of methods known as *sampling-free* approaches has been developed. These methods avoid the need to generate samples from the model distribution altogether. Instead of attempting to approximate the intractable expectation over $\rho_\theta$, they formulate surrogate objectives that are tractable and differentiable without needing the partition function $Z_\theta$. The two most prominent techniques in this family are Score Matching and Noise Contrastive Estimation, although several generalizations and related techniques exist.

**Score Matching.** Introduced by Hyvärinen & Dayan (2005), Score Matching provides a principled alternative to maximum likelihood by directly matching the scores of the model and the data. Specifically, Score Matching minimizes the squared difference between the model score $\nabla_x \log \rho_\theta(x)$ and the unknown data score, under the empirical data distribution $\rho_*(x)$. For EBMs where the model density is expressed as $\rho_\theta(x) \propto e^{-U_\theta(x)}$, we have $\nabla_x \log \rho_\theta(x) = -\nabla_x U_\theta(x)$, so the objective becomes:

$$J_{\mathrm{SM}}(\theta) = \mathbb{E}_{x \sim \rho_*} \left[ \frac{1}{2} \left\| \nabla_x U_\theta(x) \right\|^2 - \nabla_x \cdot \nabla_x U_\theta(x) \right].$$

This formulation is noteworthy for its independence from the partition function $Z_\theta$, since the score function does not involve $\log Z_\theta$. As a result, the training objective is tractable and differentiable. However, this advantage comes with a major drawback: the model does not learn a normalized density. In particular, because the loss function does not involve the log-likelihood, there is no mechanism to recover or estimate $Z_\theta$. Consequently, the learned model is suitable for tasks such as mode-seeking or denoising, but not for tasks requiring a calibrated probability density.

Moreover, the divergence term $\nabla_x \cdot \nabla_x U_\theta(x)$ (i.e., the Laplacian of the energy function) can be computationally expensive and numerically unstable, especially in high dimensions or when using deep neural network parameterizations. To alleviate this, Vincent (2011) proposed Denoising Score Matching, a variant where the score is estimated on noisy versions of the data:

$$J_{\mathrm{DSM}}(\theta) = \mathbb{E}_{x \sim \rho_*} \mathbb{E}_{\tilde{x} \sim q(\tilde{x}|x)} \left[ \left\| \nabla_{\tilde{x}} U_\theta(\tilde{x}) + \frac{\tilde{x} - x}{\sigma^2} \right\|^2 \right].$$

Here, $q(\tilde{x}|x)$ is typically taken as a Gaussian corruption kernel, and $\sigma$ is a fixed noise scale. DSM is more stable numerically and can be implemented via automatic differentiation. This formulation also plays a foundational role in the development of modern score-based generative models and diffusion models, where generative sampling is achieved via Langevin dynamics or stochastic differential equations guided by learned score functions (Song & Kingma, 2021).

**Noise Contrastive Estimation.** Proposed by Gutmann & Hyvärinen (2010), NCE formulates the training of unnormalized models as a binary classification task. The idea is to train a discriminator that distinguishes between real data samples $x \sim \rho_*(x)$ and noise samples $x \sim \rho_{\mathrm{noise}}(x)$. The EBM learns an energy function $U_\theta(x)$ such that the log-ratio $f_\theta(x) = -U_\theta(x) - \log \rho_{\mathrm{noise}}(x)$ approximates the log odds of the sample being real versus noise. The objective function is a standard logistic loss:

$$J_{\mathrm{NCE}}(\theta) = \mathbb{E}_{x \sim \rho_*} \left[ \log \sigma(f_\theta(x)) \right] + \mathbb{E}_{x \sim \rho_{\mathrm{noise}}} \left[ \log(1 - \sigma(f_\theta(x))) \right],$$

where $\sigma(z) = \frac{1}{1+e^{-z}}$ is the sigmoid function. The resemblance with the minimax formulation of GANs is clear: the energy function plays the role of a discriminator, while the noise distribution is analogous to a fixed generator.

One key advantage of NCE is that it allows for consistent estimation of the partition function $Z_\theta$, at least under certain assumptions. If the noise distribution $\rho_{\mathrm{noise}}(x)$ is known and overlaps well with $\rho_*(x)$, then $Z_\theta$ can be estimated jointly during training, as shown in Chehab et al. (2024). This contrasts with score matching methods, where the normalization constant is fundamentally unidentifiable.

However, NCE has its own limitations. Its effectiveness depends critically on the choice of the noise distribution: if the noise is too dissimilar from the data, the model receives little signal and may fail to learn meaningful structure. Conversely, if the noise is too similar, the classification task becomes too hard, leading to vanishing gradients. Moreover, in high-dimensional spaces, designing an informative noise distribution that supports the full data manifold is nontrivial, often requiring domain knowledge or adaptive methods.

**Extensions and Other Approaches.** Beyond SM and NCE, a variety of other sampling-free techniques have been explored. These include:

- **Sliced Score Matching (SSM)** (Song et al., 2020a): a variant of SM that reduces the computational cost of the trace term by projecting the gradient along random directions. This enables scaling to high-dimensional data while retaining stability.

- **Flow-Contrastive Objectives** (Gao et al., 2020): hybrid methods that use normalizing flows to approximate the model distribution or its gradients, blending the benefits of tractable likelihoods with contrastive learning. This also permits approximate estimation of the partition function via flow-based auxiliary models.

- **Denoising Score Matching with Langevin Dynamics** (Song et al., 2021): used as a building block for diffusion probabilistic models, this technique combines DSM with an iterative generative procedure based on Langevin steps or reverse-time SDEs.

Sampling-free training methods have become central to the modern landscape of Energy-Based Models, offering scalable alternatives to traditional MCMC-based maximum likelihood estimation. These approaches bypass the need to sample from the model distribution during training—a key computational bottleneck for likelihood-based learning. Despite their practical success, these methods come with fundamental limitations that constrain their applicability:

1. **Lack of normalization:** Perhaps the most critical drawback is that these methods do not yield an estimate of the partition function $Z_\theta$, except in particular cases. As a result, the learned energy function defines only an unnormalized density. This precludes the use of these models for tasks that require calibrated likelihoods or probabilistic reasoning, such as Bayesian inference, marginalization, or likelihood-based evaluation. Even approximate likelihoods are often not computable without additional assumptions or ad hoc sampling.

2. **Sensitivity to design choices:** Many contrastive methods, such as NCE and its generalizations, require the specification of a noise or negative sample distribution. The success of training depends heavily on how well this distribution matches the model's inductive biases. Poor choices can lead to degenerate or mode-collapsing solutions, and there is often no principled way to select or adapt this distribution during training.

3. **Computational challenges:** Although they avoid sampling, methods like Score Matching or its regularized and denoising variants involve computing gradients and sometimes second-order derivatives (Hessians) of the energy function. This can become prohibitively expensive in deep architectures unless approximations or tricks (e.g., Hutchinson's estimator) are used, which may further affect stability or performance.

As of today, sampling-free methods constitute a significant portion of state-of-the-art approaches to training EBMs, particularly in high-dimensional or generative contexts where MCMC is infeasible. However, because of the issues mentioned above, these methods are rarely sufficient on their own. They are often combined with auxiliary objectives, learned samplers (e.g., amortized Langevin dynamics), or score-based generative modeling frameworks (e.g., diffusion models), which reinterpret EBMs as noise-conditioned score networks. While these hybrid approaches have led to impressive results in practice, they do not resolve the fundamental limitations concerning normalization and likelihood evaluation.

Further advancements in the field have focused on improving the underlying architectures and leveraging additional tricks to alleviate some of the computational challenges and increase model flexibility. For example, the use of *energy-based deep networks* with specialized architectures like residual networks (ResNets) or transformers and convolutional layers has proven to be effective in learning complex energy functions Hoover et al. (2023). Additionally, techniques like *temperature annealing* have been used to improve convergence and prevent mode-collapse in generative settings.

In summary, while sampling-free methods are indispensable for scaling EBMs to realistic data regimes and complex architectures, they fundamentally trade away full probabilistic interpretability in favor of computational tractability. This tradeoff must be carefully considered depending on the downstream application.

## 7    Conclusion and Open Perspectives

In conclusion, Energy-Based Models represent a versatile and conceptually rich framework within the landscape of generative modeling. They provide not only powerful tools for learning data distributions but also offer deep insights grounded in the formalism of statistical physics. This review has provided a comprehensive exploration of the foundational principles and practical methodologies underlying EBMs, with a particular emphasis on their synergy with the language and techniques of statistical mechanics.

By elucidating the connections between EBMs and other generative paradigms such as Generative Adversarial Networks, Variational Autoencoders, and Normalizing Flows, we have identified the specific strengths and limitations of each approach. In particular, the centrality of energy functions and partition functions in EBM formulations mirrors the core structures of equilibrium statistical mechanics, making them uniquely suited to modeling complex systems with rich internal structure. We have also discussed the crucial role of sampling—particularly via Markov Chain Monte Carlo methods—in enabling EBMs to generate samples, approximate gradients, and estimate expectations, despite the intractability of the normalization constant in most realistic scenarios. Moreover, we have reviewed state-of-the-art training techniques for EBMs, including contrastive learning, score-based objectives, and recent sampling-free methods, along with architectural innovations and training tricks that improve stability and efficiency. These advances have helped alleviate long-standing issues such as mode collapse, vanishing gradients, and poor mixing. A central motivation of this review has been to foster deeper interdisciplinary dialogue—particularly between the communities of statistical physics and machine learning. EBMs are inherently connected to equilibrium and nonequilibrium thermodynamics through their reliance on energy-based formulations and sampling processes that parallel physical dynamics. In this light, we have argued that the integration of EBMs with Monte Carlo methods and non-equilibrium statistical mechanics is not only natural but necessary. Techniques from Langevin dynamics, thermostatted systems, and fluctuation theorems can all play a role in advancing the training and understanding of EBMs, both theoretically and computationally.

Looking forward, we identify several promising directions for future research. First, the systematic incorporation of physically inspired Monte Carlo algorithms—particularly those rooted in non-equilibrium statistical mechanics—could lead to more efficient and interpretable EBM training protocols. Second, the design of new energy functions informed by physical intuition (e.g., symmetries, conservation laws, locality) could offer inductive biases beneficial for both scientific and engineering tasks. Finally, the exploration of EBMs as physically grounded models of computation and learning opens conceptual pathways toward an interpretable and principled generative artificial intelligence. In summary, EBMs provide a robust and unifying perspective on generative modeling, with strong theoretical underpinnings and broad applicability. We hope this review serves as a valuable resource for physicists, computer scientists, and machine learning practitioners alike, helping to clarify the intricate landscape of energy-based approaches and inspiring new interdisciplinary collaborations. By emphasizing the deep connections between physics and generative modeling, we aim to contribute to a more coherent and physically motivated understanding of learning in complex systems.

## Acknowledgments

D.C. worked under the auspices of Italian National Group of Mathematical Physics (GNFM) of INdAM. D.C. expresses their gratitude to Prof. Lamberto Rondoni and prof. Eric Vanden-Eijnden for their support and the valuable discussions. D.C. gratefully acknowledges support from the Italian Ministry of University and Research (MUR) through the grant "Ecosistemi dell'innovazione", costruzione di "leader territoriali di R&S" with grant agreement no. ECS00000036. D.C. was supported in part through the NYU IT High Performance Computing resources, services, and staff expertise.

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

# A  Historical Perspective on Generative models

The problem of description of data through a mathematical model is very old, being the basis of scientific method. The set of measurements use to fulfill the role that contemporary data scientists now refer to as a dataset. Presently, the model can take the form of an exceedingly complex neural network, but the underlying extrapolation remains akin to P.S. Laplace's famous deterministic statement(Laplace, 2012): "An intellect which at a certain moment would know all forces that set nature in motion, and all positions of all items of which nature is composed [data], if this intellect were also vast enough to submit these data to analysis, it would embrace in a single formula the movements of the greatest bodies of the universe and those of the tiniest atom; for such an intellect nothing would be uncertain and the future just like the past could be present before its eyes.". One can easily extend this reasoning, asserting that the more data one possesses, the more robust and detailed the model that can be constructed atop them. This leads to enhanced predictions and greater stability concerning unforeseen behaviors.

This line of thought gained momentum in the previous century with the advent of automatic calculators, leading to an astounding pace of development. For instance, consider the remarkable difference in computational power between modern personal computers and the computer used by NASA for the Apollo program in the 1960s—the latter had only a fraction of the data processing capability available today [3]. Data has always been a fundamental part of science, but today they sort of gained the leading role as opposed to pure theory. For instance, consider the history of Kepler's laws. The genesis of such results is deeply rooted in a vast collection of observational data amassed by T. Brahe [4]. Kepler's formulation was, in fact, motivated by the necessity to explain these astronomical measurements. In a simplified analogy, we observe the dichotomy between the "model," embodied by Kepler, and the "dataset," represented in this narrative by Brahe. Since the 17th century, these two actors have played equally fundamental roles in the advancement of science, taking turns on the stage with the same importance. Consider the pivotal role played by Faraday's experiments in understanding electromagnetism (Al-Khalili, 2015), long before Maxwell's laws. Or, conversely, the impact of theory of Relativity[5] way before its experimental confirmation.

Thanks to the aforementioned technological advancements in computer science, the volume of generated scientific (and not) data has dramatically increased, resulting from advancements in simulation and storage capabilities. Thus, particularly during the 2000s, we have witnessed a profound paradigm shift represented by the Big Data Era[6] – a growing collection of data on human activities, including images, text, sounds, and more are available nowadays. This shift posed a methodological problem in what we now refer to as data science. In an historical analogy, it is akin to Brahe suddenly providing Kepler with a thousand times the amount of data that the latter was accustomed to. And this is where the machine learning approach came into play(Fradkov, 2020). As a side remark, the models required to process Big Data had already been theoretically studied since the invention of the perceptron(McCulloch & Pitts, 1943). Their application was constrained by computational power in the last century, but, as a peculiar example of convergent development, they became the primary tools in the toolbox of data scientists in the 2000s, simultaneously to the appearance of Big Data on the stage.

There is indeed a discontinuity that deserves more attention: the increasing collection of data *generated* by humans. The term *Big Data* is sometimes limited to images, sounds, videos, text, and metadata resulting from human activities. Unlike scientific measurements, having access to an extensive quantity of information produced by humans opened Pandora's box, prompting the natural question: *can we build artificial intelligence by leveraging Big Data*? In other words, can we construct a machine capable of *generating* data as humans do, by training it in some smart way? Data here is to be understood in a broad sense, encompassing new theorems, art pieces, images, videos, and even novels.

Generative models represent, in this sense, the most recent breakthrough in technological advancement towards intelligent-like machines. It is complicated to provide a general definition, and there are already many available from different sources[7]. However, if we informally focus on those already known to the general public, such as Generative Pre-Trained Transformers (GPT)[8], a possible simple definition is: *generative*

---

[3]https://www.linkedin.com/pulse/smartphone-today-has-more-computing-power-than-nasas-1960-offermann
[4]https://www.britannica.com/science/history-of-science/Tycho-Kepler-and-Galileo
[5]https://www.britannica.com/science/relativity/Intellectual-and-cultural-impact-of-relativity
[6]https://medium.com/swlh/big-data-era-84b488491a8d
[7]https://www.nvidia.com/en-us/glossary/generative-ai/
[8]https://www.nytimes.com/2022/12/10/technology/ai-chat-bot-chatgpt.html

*models model the observed data through a probabilistic program, able to output samples from the distributions.* Returning to the historical analogy: nowadays, we are able to build "BraheGPT", which can generate and gather new plausible measurements about the orbits of planets in unobserved planetary systems after training on observed data from the solar system. However, it is not Kepler – the probabilistic model is generally not informative about the dataset; deductive reasoning is not necessary to generate new data instances, although it remains fundamental to understanding the world. Von Neumann would certainly adapt his famous statement(Dyson et al., 2004) about overfitting to modern data science, cautioning against the ability to generate examples without a general picture.

Prominent data scientists, such as Yann LeCun, have recently emphasized that the use of interpretable generative models is crucial for achieving a "unified world model for AI capable of planning"[9]. This thesis becomes imperative in the realm of computational sciences, where qualitative generation alone is insufficient as a benchmark to evaluate model performance. In sectors like Molecular Dynamics, Biochemistry, and similar fields, the model must convey substantial information about the dataset. The generative models that excel in terms of interpretability, which form the main focus of the present work, are precisely the *Energy-Based Models* (EBMs). These models offer a unique advantage in their ability to provide insights into the underlying mechanisms of the data they generate. The reason is that a trained EBM provides a *normalized* probability model – having such object is equivalent to interpretability since you can give a probabilistic meaning to each sample, generated and not. One can exploit this fact for instance in anomaly detection (Zhai et al., 2016). On the contrary the original formulation of state-of-the-art methods as flow-based ones do not output a normalized probability, but usually an efficient sampling method. For this reason building a link between EBMs and other generative models is a very active line of research (Che et al., 2020)(Chao et al., 2024). In areas such as Molecular Dynamics and Biochemistry, where understanding the intricate relationships within the dataset is crucial, the interpretability of EBMs stands out. As the pursuit of a unified world model for AI continues, the emphasis on interpretable generative models, particularly EBMs, plays a pivotal role in bridging the gap between data generation and comprehensive understanding.

### A.1 A long story: from Boltzmann-Gibbs ensemble to the advent of EBMs

After outlining the historical evolution of generative models, this section explores the origins and development of Energy-Based Models. The theoretical foundation of EBMs spans multiple disciplines, including statistical physics, probability theory, and computer science, often appearing under different names. Here, we adopt a historical perspective, while deeper theoretical discussions are deferred to later chapters. This review aims to provide a broad audience with a coherent understanding of EBMs by contextualizing their emergence.

Since an EBM is a probabilistic model, it is natural to ask for its definition. To answer, we begin with the Boltzmann-Gibbs measure, a fundamental concept in statistical mechanics originating from the works of Ludwig Boltzmann and Josiah Willard Gibbs in the late 19th century. Boltzmann introduced the statistical interpretation of entropy and the Boltzmann distribution (Boltzmann, 1868), linking microscopic configurations to macroscopic thermodynamics. In parallel, Gibbs formalized these ideas into the canonical ensemble and the Gibbs measure (Gibbs, 1902), providing a mathematical framework for thermodynamic properties. The resulting Boltzmann-Gibbs measure describes the probability distribution over energy states at *thermal equilibrium* at temperature $T$, forming the foundation of statistical mechanics across diverse physical systems. We informally recall its definition: given the state of the system $x \in \Omega$, where $\Omega$ is the so-called phase space, and an energy function $U : \Omega \to \mathbb{R}^+$, we can express the associated probability density function

$$\rho(x) \propto e^{-\beta U(x)} \tag{107}$$

where $\beta = 1/k_B T$, $k_B$ being the Boltzmann constant. A detailed mathematical description will be provided in the next chapters.

The next chapter of the story happens in 1924, when E. Ising presented his PhD thesis[10]. The so called Ising model is a fundamental mathematical model in statistical mechanics. It serves as a simplified yet powerful representation of magnetic systems, which the Ising Model represents as a graph with N nodes. In the Ising

---

[9]https://www.zdnet.com/article/metas-ai-luminary-lecun-explores-deep-learnings-energy-frontier/

[10]https://www.hs-augsburg.de/~harsch/anglica/Chronology/20thC/Ising/isi_fm00.html

model, each lattice site is associated with a magnetic spin, which can take two possible values: +1 ("up") and -1 ("down")", yielding $\Omega = \{-1, 1\}^N$ as set of possible lattice configurations. The energy of the system is modeled as

$$U_{Is}(x) = -\sum_{\langle i,j \rangle} J_{ij} x_i x_j - \mu \sum_j h_j x_j \tag{108}$$

where $i, j \in \Lambda$ are indexes of sites in the lattice; $\langle i, j \rangle$ is the standard notation (Cipra, 1987) to denote that the sum is restricted to first neighbours and $J_{ij}$ is the strength of the interaction. The field $h_i$ instead individually acts on each site and $\mu$ is just a constant that traditionally corresponds to magnetic moment. In laymen terms, each magnetic spin interacts with its first neighbours and with an external field. The alignment of spins is encouraged.

In considering equation 107 as associated to $U_{Is}$, the primary focus is often on the behavior of the system as a function of temperature. In a nutshell, at high temperatures, thermal fluctuations dominate, and the system exhibits no long-range order, i.e. spin assumes up and down state with equal probability. As the temperature decreases, there is a critical point at which the system undergoes a phase transition, leading to spontaneous "ordering", i.e. magnetizaion, of the system, that is either most of the spin are up or down respectively.

For some decades the interest for Ising model and its extensions was confined to physics. The motivation for invoking such a model in the present work is the following: in the 80s a fundamental connection between Ising model and data science manifested through Hopfield networks(Hopfield, 1982) and Boltzmann machines(Ackley et al., 1985). Both can be viewed as an Ising lattice where interactions are not confined to first neighbors. Apart from the initial summation, which, for the former, extends to $\forall i, j \in \Lambda$ rather than just $\langle ij \rangle$, the energy function bears resemblance to equation 108. From a statistical physics standpoint, the distinction between a Hopfield network and Boltzmann machines lies solely in the temperature value.

The purpose of the former is pattern recognition and associative memory tasks. A distinctive feature of Hopfield networks is their proficiency in storing and retrieving patterns through symmetric connections between neurons, that is Ising sites, in the network. In practice, when provided with a set of network configurations $y^\lambda \in \Omega$ representing patterns, denoted by $\lambda = 1, \ldots, n$, one constructs the coupling $J$ as follows:

$$J_{ij} = \frac{1}{n} \sum_{\lambda=1}^{n} y_i^\lambda y_j^\lambda \tag{109}$$

This involves employing the Hebbian rule(Hebb, 1949) "neurons wire together if they fire together"(Löwel & Singer, 1992), but further specifications(Storkey, 1997) are available. This phase is commonly referred to as the *training* of the network. Subsequently, one can define a retrieval iterative dynamics starting from any configuration $x^{k=0} \in \Omega$, as exemplified by the equation:

$$x^{k+1} = \text{sgn}(Jx^k + h) \qquad k \in \mathbb{N} \tag{110}$$

Here, $J$ represents the coupling matrix defined element-wise in equation 109, and $h$ is a bias vector that influences the preferences for 'up' or 'down'. It is noteworthy that in a Hopfield network, there is no use of the Boltzmann-Gibbs ensemble; the objective is to construct a dynamical system with prescribed attractors, which are the minima of $U(x)$ by design.

Boltzmann Machines share the same structure and energy function but the goal extends beyond the mere retrieval of patterns; it is to model their overall *distribution*. To illustrate this concept, consider a finite set of $n$ natural images of cats and dogs. A meticulously designed Hopfield Network could perfectly retrieve any of these examples. On the contrary, a trained Boltzmann Machine aspires to generate new instances of cats and dogs, capturing, in a sense, the distribution of such images. The objective appears to be on a different level of difficulty: although possibly big, the cardinality of the set of patterns is finite; the number of possible variations of cats and dogs is not. Thus, one can immediately guess why the training and generation phases (n.b. it is no more just a retrieval) are completely different w.r.t. Hopfield Networks. The take home message is the hypothesis that the distribution of the given patterns can be described by a Boltzmann-Gibbs ensemble associated to the energy of the Hopfield Network at temperature $T$.

It is convenient to consider Boltzmann Machines as a specific instance of Energy-Based Models, a term introduced by Hinton et al. (Teh et al., 2003), to describe both training and generation phases. EBMs

differ from Boltzmann Machines in the use of a generic parametric energy $U_\theta(x)$ instead of the usual choice made for the latter. Here, $\theta \in \Theta$ needs to be selected and trained so that the Boltzmann-Gibbs ensemble $\rho_\theta$ associated with $U_\theta(x)$ "fits well" the distribution of the given patterns, which we refer to as $\rho_*$. After training the EBM, the generative phase involves *sampling* equilibrium configurations from $\rho_\theta$. Specifically, a Boltzmann Machine corresponds to an EBM with the choice of $U(x)$ as the energy of a Hopfield Network and $\theta = J$.

Despite their conceptual simplicity, both training and generation represent fundamental open problems that intersect multiple research fields. In essence, sampling from a Boltzmann-Gibbs ensemble is a challenging task in general, and unfortunately, it is necessary even during the training phase. For this reason, the use of Boltzmann Machines was limited to toy models until the proposal of the Contrastive Divergence algorithm by Hinton (Hinton, 2002).

This procedure, along with its generalizations, made it possible to apply EBMs to practical problems. Moreover, thanks to the adoption of a deep neural network (Xie et al., 2016) as $U_\theta$, the interest towards this class of generative models critically increased and in 2010s the use of EBMs for state-of-the-art tasks became standard. However, all that glitters is not gold. Despite its success in generating high-quality individual samples, the use of Contrastive Divergence is known to be biased. For instance, it could happen that individual images are correctly generated, but ensemble properties as the relative proportion of the two species is incorrect. Although Hinton et al. originally claimed that this bias is generally small (Carreira-Perpinan & Hinton, 2005), numerous counterexamples have been shown in the more than 20 years since their original paper. The absence of novel paradigm shifts, coupled with the rise of alternative generative models (e.g., diffusion-based ones (Song et al., 2020b)), has reduced attention on EBMs and consequently on Boltzmann Machines. Nevertheless, recent advances in sampling algorithms, optimization techniques, and theoretical understanding—particularly from the perspective of statistical physics—have sparked a renewed interest in EBMs. Their interpretability, connection with energy landscapes, and potential for modeling structured and multi-modal data suggest that EBMs still hold substantial promise in modern generative modeling.

## B   Stochastic Interpolants

**Definition B.1.** *Given two probability densities $\rho_1, \rho_2 : \mathbb{R}^d \to \mathbb{R}_+$, a* **stochastic interpolant** *between them is a stochastic process $X_t \in \mathbb{R}^d$ such that*

$$X_t = I(t, X_0, X_1) + \gamma(t)z \qquad t \in [0, 1] \tag{111}$$

*where:*

- *The function $I$ is of class $C^2$ on its domain and satisfy the following endpoint conditions*

$$I(i, X_0, X_1) = X_i \qquad i = 0, 1 \tag{112}$$

  *as well as*

$$\exists C_1 < \infty \: : \: |\partial_t I(t, X_0, X_1)| \le C_1 |X_0 - X_1| \quad \forall (t, X_0, X_1) \in [0, 1] \times \mathbb{R}^d \times \mathbb{R}^d \tag{113}$$

- *$\gamma : [0, 1] \to \mathbb{R}$ is such that $\gamma(0) = \gamma(1) = 0$ and $\gamma(t) > 0$ for $t \in (0, 1)$.*

- *The pair $(X_0, X_1)$ is sampled from a measure $\nu$ that marginalizes on $\rho_0$ and $\rho_1$, that is $\nu(dX_0, \mathbb{R}^d) = \rho_0 dX_0$ and $\nu(\mathbb{R}^d, dX_1) = \rho_1 dX_1$.*

- *The variable $z$ is a Gaussian random variable independent from $(X_0, X_1)$, i.e. $z \sim \mathcal{N}(\mathbf{0}_d, \mathbf{I}_d)$ and $z \perp (X_0, X_1)$*

Let us focus on the case in which $\rho_0 = \bar{\rho}$ to be a simple base distribution (e.g. a Gaussian) and $\rho_1 = \rho_*$, that is the data distribution. Equation equation 111 means that if we sample a couple $X_0 \sim \rho_0$ and $X_1$ from the dataset, the interpolant is a stochastic process that connects the two points. The objective is to build a generative model that, in some sense, learns from the interpolants the way to map samples from $\bar{\rho}$ to $\rho_*$. The first important result in this sense is the following(Albergo & Vanden-Eijnden, 2022):

**Proposition B.1.** *The interpolant $X_t$ is distributed at any time $t \in [0, 1]$ following a time dependent density $\rho(x, t)$ such that $\rho(x, 0) = \rho_0$ and $\rho(x, 1) = \rho_1$, and also satisfies the following transport equation:*

$$\partial_t \rho + \nabla \cdot (b\rho) = 0 \tag{114}$$

*where the vector field $b(x, t)$ is defined by a conditional expectation:*

$$b(x, t) = \mathbb{E}\left[\dot{X}_t \mid X_t = x\right] = \mathbb{E}\left[\partial_t I\left(t, X_0, X_1\right) + \dot{\gamma}(t)z \mid X_t = X\right] \tag{115}$$

*Proof.* Let $g(k, t) = \mathbb{E}e^{ik \cdot X_t}$ the characteristic function of $\rho(x, t)$, that is

$$g(k, t) = \mathbb{E}e^{ik \cdot (I(t, X_0, X_1) + \gamma(t)z)} \tag{116}$$

If we compute the time derivative of $g$, we obtain

$$\partial_t g(k, t) = ik \cdot m(k, t) \tag{117}$$

where $m(k, t) = \mathbb{E}[(\partial_t I\left(t, X_0, X_1\right) + \dot{\gamma}(t)z)e^{ik \cdot X_t}]$. By definition of conditional expectation,

$$\begin{aligned} m(k, t) &= \int_{\mathbb{R}^d} \mathbb{E}[(\partial_t I\left(t, X_0, X_1\right) + \dot{\gamma}(t)z)e^{ik \cdot X_t} \mid X_t = x]\rho(x, t)dx \\ &= \int_{\mathbb{R}^d} e^{ik \cdot x}b(x, t)\rho(x, t)dx \end{aligned} \tag{118}$$

where we used the definition of $b$. If we insert $m(k, t)$ in equation 117 and we compute the Fourier anti-transform, we immediately obtain equation 114 in real space. $\square$

Other properties of $b$ can be proven, but for the sake of the present summary we will not delve into them. As usual we can identify $\bar{\rho}$ and $\rho_*$ as base and data distributions. Thanks to the previous Proposition we can already define a diffusion-based generative model:

**Lemma B.1** (ODE Generative Model). *Given Proposition B.1 and $\rho(x, 0) = \bar{\rho}$, the choice $\mu(X_t, t) = b(X_t, t)$ and $\sigma(X_t, t) = 0$ in equation 40 satisfies the endpoint condition for $T = 1$.*

Differently from score-based diffusion models, such ODE-based formulation does not involve stochasticity during generation. In fact, the ODE $\dot{X}_t = b(X_t, t)$ can be interpreted as a Normalizing Flow (see Section 6) where the pushforward is defined via a transport PDE. Interestingly, the ODE formulation is formally equivalent to an SDE formulation:

**Lemma B.2** (SDE Generative Model). *For $\varepsilon > 0$, given Proposition B.1, $\rho(x, 0) = \bar{\rho}$ and the score $s(x, t) = \nabla \log \rho(x, t)$, the choice $\mu(X_t, t) = b(X_t, t) + \varepsilon s(X_t, t)$ and $\sigma(X_t, t) = \sqrt{2\varepsilon}$ in equation 40 satisfies the endpoint condition for $T = 1$.*

*Proof.* Adding and subtracting the score to equation 114, we obtain for any $\varepsilon > 0$

$$\partial_t \rho + \nabla \cdot ((b + \varepsilon s - \varepsilon s)\rho) = 0 \tag{119}$$

But $s\rho = \nabla \rho$, that is

$$\partial_t \rho + \nabla \cdot ((b + \varepsilon s)\rho - \varepsilon \nabla \rho) = 0 \tag{120}$$

Trivially, the solution of the PDE equation 120 is the law of a stochastic process solution of an SDE as in equation 40. $\square$

We presented the proof as an example of the standard trick used to convert the diffusion term into a transport term exploiting the score.

We defined the generative model, but similarly to score-based diffusion, we need to clarify how $b$ and $s$ are learned in practice from data. For such purpose, we present the following result:

**Proposition B.2.** *The vector field $b(x, t)$ is the unique minimizer of the following objective loss*

$$\mathcal{L}_b[\hat{b}] = \int_0^1 \mathbb{E} \left( \frac{1}{2} \left| \hat{b}\,(t, X_t) \right|^2 - (\partial_t I\,(t, X_0, X_1) + \dot{\gamma}(t)z) \cdot \hat{b}\,(t, X_t) \right) dt \tag{121}$$

*Similarly, the the score $s(x, t)$ the unique minimizer of the following objective loss*

$$\mathcal{L}_s[\hat{s}] = \int_0^1 \mathbb{E} \left( \frac{1}{2} \left| \hat{s}\,(t, X_t) \right|^2 + \gamma^{-1}(t)z \cdot \hat{s}\,(t, X_t) \right) dt \tag{122}$$

For the sake of the present summary, we will not present the proof(Albergo & Vanden-Eijnden, 2022). The take home message is that one can now propose two neural networks, namely $b_\theta(x, t)$ and $s_{\theta'}(x, t)$, and train them through backpropagation using equation 121 and equation 122. The integrals are estimated using random pairs $(X_0, X_1) \sim \nu$ and times $t \sim \mathcal{U}[0, 1]$. As for score-based diffusion, we avoid delving into practical details regarding the implementation of the neural networks. We emphasize the main message: it is feasible to construct a diffusion model defined in a finite time interval that does not solely rely on the score function. In fact, score-based diffusion can be viewed as a specific instance of stochastic interpolation or similar methods (refer to Section 3.5 for more details).

Concerning practical aspects, the freedom in choosing the function $I(t, X_0, X_1)$ as well as $\gamma(t)$ can be challenging due to the absence of a general guiding principle. Unfortunately, the structure of the interpolant and the implementation of $b_\theta$ and $s_{\theta'}$ can significantly impact the efficient training of the model. Regarding the generative phase, the SDE and ODE formulations are formally equivalent, but the practical choice is not straightforward. From a numerical perspective, the primary issue lies in the time discretization and integration of the differential equations. The ODE is preferred since integration methods are more stable and precise compared to those for SDEs; this allows for larger time steps and accelerated generation. This is also a significant advantage of stochastic interpolants over score-based diffusion, which is SDE-based. However, the presence of noise appears to be necessary as regularization: in layman's terms, since $b$ is learned and possibly imperfect, any mismatch is "smoothed" in the SDE setting by the presence of noise. The value of $\varepsilon$ functions as a hyperparameter in this context.

