# OpenReview forum: "Hitchhiker's guide on the relation of Energy-Based Models with other generative models, sampling and statistical physics: a comprehensive review"
_TMLR — Accepted by TMLR_

### Review · Reviewer_izai · 2024-12-14

**Summary Of Contributions:**

This paper presents a comprehensive review of Energy-Based Models (EBMs) and their relation to other generative models and statistical mechanics principles. The paper discusses various EBM training and sampling techniques and highlights the parallels between EBM concepts and statistical mechanics.

**Audience:**

No

**Broader Impact Concerns:**

There is no ethical concerns.

**Claims And Evidence:**

No

**Requested Changes:**

I think key insights and key results, as well as practical implications needed to be better highlighted. In the current form I felt the paper is not particularly helpful for a researcher trying to learn about EBMs -- what they are good for, when do they fail, how do they connect to other state-of-the-art techniques.

**Strengths And Weaknesses:**

This paper is a survey/review paper. This reviewer is not exactly familiar the criterions for reviewing an review paper, but felt the paper is overall a bit unorganized and it is unclear what to take away from it. To be specific:

Strength:

- The paper is quite comprehensive in the materials it presents as it covers many aspects of EBMs.

Weaknesses:

- The text in the paper is mainly about general information about EBMs and other types of generative models. It is not very dense in the state-of-the-art approaches in EBMs, or the connections between EBMs and other types of generative models.
- This reviewer wasn't able to take away key insights from the paper. This reviewer felt a lot of the content are more like a tutorial or general introduction to the topic, which may be found in existing literature.
- With a lot of general information, the paper is rather lenghty, of 41 pages in general. The paper is aimed to be a "Hitchhiker’s guide on Energy-Based Models". However,  I felt the information flow is not very condensed or easy to follow.
- Some definitions and theorems are not stated in an organized and rigorous way.

---

> ### Author Response · Authors · 2024-12-19
>
> We thank you the reviewer for the careful analysis of our work and for the highlighted strengths and weaknesses. We will try to address the latter by points in this reply, in order to propose possible revisions that matches your requested changes.
>
> Reply to weaknesses:
>
> -We acknowledge that the paper covers a broad range of topics in Energy-Based Models (EBMs), which makes it more general in nature. This is intentional: our aim is to provide a comprehensive but accessible resource that serves both as an introduction for newcomers, and a synthesis of ideas that might reveal unnoticed connections to experienced researchers. Diving deeply into any single subtopic would make the paper excessively lengthy, making it closer to a book, and detract from its purpose as a broad overview.
>
> -We appreciate the reviewer's suggestion to better highlight key insights, results, and practical implications. We will revise the content to ensure that the takeaways for both beginners and experts are more explicitly stated. This includes summarizing the practical strengths, limitations, and use cases of EBMs in clearly marked sections. While the paper touches on these aspects, we will incorporate additional content to specifically address the latest advances and clarify the connections between EBMs and other generative models. This will help bridge the gap between foundational concepts and state-of-the-art techniques.
>
> -We recognize that the paper is lengthy due to its comprehensive scope. We tried to balance clearness of contents with a sufficiently deep analysis. To improve readability and flow, we will introduce a clearer organizational structure, with signposted sections summarizing key points and transitions. This should help readers quickly locate the most relevant information based on their interests. Regarding the title, given your suggestion we agree that we could have possibly chosen a different one, for instance "Hitchhiker's guide on the relation of Energy-Based Models with other generative models, sampling and statistical physics: a comprehensive review" given that the accent of the whole treatment should be the relation with other topics more than EBMs themselves.
>
> -Our aim with definitions and theorems was to provide a general understanding of the tools and results behind each topic. A full mathematical treatment is presented only when it aligns with the paper's length constraints. For results requiring significant detail, we opted to present simplified versions of milestone theorems and provide references for readers seeking more rigorous analysis. To address this concern, we propose explicitly stating this approach in the introduction. Additionally, we will refine and correct any oversimplified sections that may benefit from greater precision, based on your specific feedback.

---

### Review · Reviewer_BJq4 · 2024-12-19

**Summary Of Contributions:**

This manuscript contains a description of energy-based models (EBMs) and related physical systems. Additionally, it describes other modern probabilistic models, outlines some points of comparison with EBMs, and describes some EBM training and sampling algorithms.

After an introduction highlighting the links between data collection and modelling improvements, the manuscript follows by a review on the probabilistic description of physical systems based on their energy, which inspired EBMs.
Following that, a formal definition of EBMs is given in Section 2.

Section 3 introduces alternative modern probabilistic modelling approaches ---namely Variational Autoencoders, Normalizing Flows, Generative Adverserial Networks and Diffusion Models, and---briefly---compares them to EBMs.

Section 4 is a 5-page primer on basic Monte Carlo Markov Chain techniques used in EBMs, including Metropolis Hasting (MH) Algorithms and the Unadjusted---and Metropolis-Adjusted---Langevin Algorithm (ULA, MALA).

Section 5 (3 pages) contains an introduction of the links between Boltzman-Gibbs distributions and Maximum Entropy.

Finally, Section 6 reviews the Contrastive Divergence algorithm, a training algorithm used to review Energy-Based Models.

**Audience:**

No

**Broader Impact Concerns:**

None.

**Claims And Evidence:**

No

**Requested Changes:**

What I would expect, instead, from a review paper on the the relation between EBMs and other generative models is a comprehensive comparison of theoretical and practical insights of the main generative models established by the machine learning community, with as particular focus on EBMs.

- On the theoretical side, I would expect a review on known results regarding some important properties (approximation properties, sample efficiency, ability to handle manifold data, etc.) of some generative models, and whether or not these hold for EBMs, and how do such result compare to analogue results established for other generative models. I would expect formally restating some important results, not only when describing the models, but actually when describing their properties.
- On the experimental side, I would expect a thorough literature review on works having used EBMs for various applications in recent years, showcasing in which instances they have been a success. A nice touch would be a particular focus on physics/chemistry/thermodynamics applications, given the physics inspiration of EBMs.

**These, in my opinion, form key aspects of a review paper on "a comprehensive review on the relation with other generative models, sampling and statistical physics", which need to take up a significant part of the its real estate, and are absent from the current submission.**



Additionally, there are some less fundamental, but still important issues with the paper:

- I find the paper to be unnecessarily wordy in places. Conciseness is important to not waste the readers's time, and I encourage the authors to try to systematically reduce redundancy, wordiness, and analogies which sometimes do not add enough to justify the time spent introducing them.
- Some mathematical results are not properly stated (Lemma 5.1 for instance: "Maximum Entropy principle" under which constraints?) "Constrained minimization of energy functional" under which constraints?).
- I appreciate that the focus on CD due to its popularity and connection with sampling. I find it odd to not spend a few paragraphs on score matching and noise contrastive estimation, which are two other popular methods for training EBMs, and arguably simpler to use in practice.
- I understand that review papers can have a broad introduction and focus on approachability to a large audience, however I believe that this manuscript has sacrificed too much conciseness, precision, clarity and scientific rigor in the process. Below is a thorough list of concerns related to the introduction.

	1.1 Generative models

	- "An intellect which at a certain moment would know all forces [data] that set nature in motion [...] would be uncertain and the future just like the past could be present before its eyes": the way the quote is reduced is very confusing. Add "nothing" before "uncertain".
	- "This line of thought was boosted in the previous century with the advent of automatic calculators, and the velocity of development becomes astounding": syntactically odd sentence.
	- "For instance, consider the remarkable computational power difference between your smartphone and the computer used for the Apollo program by NASA in the 1960s": probably better to explain which is better explicitly. Also, let's not assume the reader has a smartphone.
	- Hence, the quest for data has become an indispensable aspect of contemporary science: logical link is not completely obvious.
	- "To delve deeper into this issue, let us construct a historical metaphor": not a metaphor, rather an analogy.
	- Non-conventional combination of line breaks and newlines.
	- "In recent years, particularly during the 2000s, we have witnessed a profound paradigm shift represented by the Big Data Era. Thanks to the aforementioned technological advancements in computer science, the volume of generated scientific (and not) data has dramatically increased, resulting from advancements in simulation and storage capabilities. Furthermore, there has been a growing collection of data on human activities, including images, text, sounds, and more". This whole cluster of paragraph needs some restructuring. the logical flow gets very interrupted, there's a lot of jumping back and forth.
	- "Returning to the historical analogy, it is akin to Brahe suddenly providing Kepler with a thousand times the amount of data that the latter was accustomed to. This shift posed a methodological problem in what we now refer to as data science". The second sentence relates to the previous paragraph. It's odd to have it bundled with the first sentence which is unrelated.
	- "not just on the internet. " Don't see what this brings to the point. Also, the internet was not mentioned before, its not clear why the sentence deserves a clause like this.
	- "Generative models represent, in this sense, the most recent breakthrough in technological advancement towards intelligent-like machines. It is complicated to provide a general definition, and there are already many available from different sources5. However, if we informally focus on those already known to the general public, such as Generative Pre-Trained Transformers (GPT)6, the common traits of most definitions are few. Firstly, generative models require a substantial amount of data for training, in addition to the selection of a precise architecture, which goes far beyond the original perceptron. Secondly, the training is probably not biologically inspired, i.e., we do not learn through backpropagationGrossberg (1987), which is the most commonly used training technique in machine learning. For completeness, it is worth noting that this thesis is still debated in neuroscienceLillicrap et al. (2020). Thirdly, a generative model is not necessarily informative about the data distribution; for instance, ChatGPT could achieve astounding results in text generation, but the training machine does not provide knowledge about some general features of text generated by humans". How about: "generative models model the observed data through a probabilistic program, able to output samples from the distributions"? The GPT-class of models, being autoregressive, fit that definition. I don't find the "training not biologically inspired" or "large training set" aspects relevant to the definition. For that reason, I'm not a big fan of placing that much importance on "generative modelling" as IMO, EBMs are not generative models per se, they are **probabilistic models** (e.g. models of the probability of some random variable). They don't define a generative process.
	- "For completeness, it is worth noting that this thesis is still debated in neuroscienceLillicrap et al. (2020). " this probably should be put in a footnote to not disturb the flow.
	- "Thirdly, a generative model is not necessarily informative about the data distribution; for instance, ChatGPT could achieve astounding results in text generation, but the training machine does not provide knowledge about some general features of text generated by humans.". True, but a negative definition, so not a super strong point to make when defining generative models.
	- "BraheGPT,"  -> "BraheGPT",
	- "The generative models that excel in terms of interpretability, which form the main focus of the present work, are precisely the Energy- Based Models (EBMs). These models offer a unique advantage in their ability to provide insights into the underlying mechanisms of the data they generate." I think the arguments in favor of the interpretability properties of EBMs should be clearly explained. Right now, this sentence is not backed up.
	- "In adopting EBMs, researchers and practitioners gain not only the capacity to generate high-quality data but also a clearer understanding of the factors influencing the generated outputs. This interpretability is indispensable in domains where the model’s ability to convey meaningful information about the dataset is paramount" lots of overlap with the previous paragraph.


	1.2

	- The first ingredient of the story: does not really motivate how this ``ingredient'''will matter. Actually, by the end of this paragraph (after equation 1), we still don't get why Boltzmann and Gibbs contributions are relevant to EBMs.
	- "The analysis of the impact of Boltzmann-Gibbs ensemble on physics would require a full monography per se; for the sake of the present work" this is not very concise. I'd suggest to just remove this sentence.
	-  "particularly in understanding the behavior of spins in a lattice Λ — for simplicity, we can imagine a graph with N nodes" -> "which the Ising Model represents as a graph with N nodes".
	- Maybe "usually denoted as "up" or "down""  -> "... two possible values: +1 ("up") and -1 ("down")", yielding a $\Omega = \{ -1, 1\}^N$ as set of possible lattice configurations.
	- "The interactions between spins are typically modeled using a simple energy function, namely" -> "The energy of the system is modeled as": ...
	- "Let us briefly clarify the notation:" -> "where" (for conciseness).
	- <ij>: is this standard notation? I've never seen it. Otherwise, I would advise the authors use standard notation especially that <...> is sometimes used to denote average in physics.
	- "In considering equation 1 as associated to UIs": not the most elegant formulation
	- "magnetization" and "long-range orders" are not defined, so the sentences mentioning it feel a bit obscure.

	Very minor:
	- Typo: This variational formulation identify  -> "identifies".
	- Please use \citep to cite papers.
	- "monography": monograph is more standard I believe. (section 1.2)

**Strengths And Weaknesses:**

**Strengths** Most of the technical content is correct/rigorously stated. An emphasis is made on links to physical systems (spin lattices, thermodynamic gases etc.), which is helpful to understand links between energy and probability and demystify the term Energy-Based Models. Section 6 contains some recent insights on CD which are not as known as they should be and it is nice to see a paper mentioning them.


**Weaknesses**: At a high, level, I don't think the paper lives up to what the title and abstract promise. The paper spends the majority of its page count introducing related concepts (other probabilistic models, maximum entropy principle, sampling techniques). The little space actually detailing their relation to EBMs contains straightforward, high-level content which is far from enough to impose the reader to read such a long background. More specifically:

- Regarding connections between EBMs and other probabilistic models, the paper currently spends nearly 50% of its total page count on introducing other models, and discussing the connection with EBMs in 4 bullets points, which I found to be too straightforward to be informative.
- Similarly, regarding connections between EBMs and statistical physics, the paper spends multiple pages introducing the maximum entropy principle, but the link to EBMs is limited to
  - one Lemma which attempts as a link between EBMs and Entropy which relies on an unmotivated identification (F = - \log Z),
  - A few high level sentences attempting to relate the Maximum Entropy Principle to training and sampling in EBM at a very high level, which I personally don't understand, and in any case will have at most as much value as two high-level sentences may have. Overall, I fail to see what the maximum entropy principle brings to the table for EBMs.
- The sampling parts virtually contains only standard textbook content on MH/MALA/ULA.

---

> ### Author Response · Authors · 2025-02-04
>
> We sincerely appreciate the reviewer’s thoughtful feedback and constructive suggestions. Their comments have provided valuable insights, allowing us to refine our manuscript and improve its clarity and balance.
>
>
>
> The aim of this paper is to review the relationships between Energy-Based Models (EBMs), generative models, sampling, and statistical physics. While reviews on these individual topics exist, we believe that presenting a structured and cohesive perspective on their interplay offers a novel contribution. Our approach has been to provide the necessary background in a rigorous yet concise manner, prioritizing self-containment over extensive external referencing. This stylistic choice is intended to make the review accessible to a broad audience, from experts in generative models who may find statistical physics unfamiliar to statistical physicists approaching EBMs for the first time.
>
>
>
> We fully acknowledge the reviewer’s concerns regarding the balance between EBMs and complementary methods, as well as the level of pedagogical exposition. Indeed, we recognize that some sections may appear too detailed for certain readers, while others may feel that the emphasis on EBMs could be stronger. We greatly appreciate these observations, as they will help us fine-tune the manuscript to better align with its intended scope and audience.
>
>
>
> Our hope is that this review will serve as both a useful reference and a source of new perspectives. If, after reading it, a reader were to think,  _“This is clear, this as well… but this connection—I had never considered it before,”_  we would consider it a great success.
>
>
>
> With this in mind, we now address the specific points raised by the reviewer.
>
> - We appreciate the reviewer’s observation regarding the balance between the introduction of other models and the discussion of their connections to EBMs. We recognize that the section outlining these relationships could benefit from a more in-depth treatment to enhance its informativeness. In response to this, and in alignment with the comments raised in the  _Requested Changes_, we will expand this section by incorporating a more detailed discussion of fundamental properties that EBMs share—or do not share—with other generative models. While we will ensure that the section remains concise and does not become overly extensive, we will increase the number of explicit references to key similarities and differences, making the connections clearer and more structured.
> - We appreciate the reviewer’s critical perspective on the role of the maximum entropy principle in our discussion. Our intention in introducing it was to present a concise yet rigorous derivation of the Boltzmann distribution, which is the main focus of Section 5, as explicitly stated at its beginning. Among the possible approaches to derive this result, we opted for the maximum entropy principle because, in our view, it allows for a mathematically precise treatment without becoming overly verbose. The goal is to provide a clear justification for the structure of the probability density function that is used in EBMs.  Regarding the specific points raised: for  _Lemma 5.1_  and the identification of  F = -\log Z , we will include external references that support and contextualize these statements, making their motivation clearer. As for the role of the maximum entropy principle, we acknowledge that its connection to EBMs could be made more explicit. To this end, we will also highlight at the end of Section 5 that this derivation is meant to provide the reader with a compact mathematical introduction to the Boltzmann-Gibbs distribution, incorporating key concepts such as free energy and partition functions. These elements recur frequently in the discussion of EBMs, and we believe this approach offers a useful bridge between statistical physics and the formalism of EBMs.
> - As stressed above, our goal with this review is to make it as self-contained as possible while remaining accessible to the broadest possible audience of researchers interested in EBMs. Given this, we felt it was essential to include a pedagogical discussion of sampling techniques in the specific context of EBMs. In particular, our experience suggests that many practitioners working with EBMs are often agnostic about certain fundamental subtleties in sampling, despite its crucial role in both training and inference. By explicitly presenting these methods within the EBM framework, we aim to provide a solid conceptual foundation that can benefit both newcomers to the field and experts who may not have encountered these details in a systematic way.
>
> (Continue below)

---

> > ### Author Response · Authors · 2025-02-04
> >
> > Regarding Requested Changes:
> > - we sincerely appreciate the reviewer’s thoughtful feedback on the scope of the review and agree that a comparative discussion of EBMs with other generative models, both from a theoretical and practical perspective, is essential. In particular, we recognize the importance of expanding the section on the fundamental properties of generative models as opposed to EBMs. We will revise this section to ensure that key properties such as approximation capabilities, sample efficiency, and the ability to model data on manifolds are discussed more systematically in relation to EBMs.
> > At the same time, we believe that a comprehensive review on this topic must also include a clear and self-contained introduction to generative models, sampling techniques, and their statistical physics connections, rather than relying on extensive external references. Our intention is to make the review accessible to a broad audience, from experts in generative models who may be less familiar with statistical physics to physicists approaching EBMs for the first time. For this reason, while we will expand the comparative discussion, we believe it is equally important to maintain the sections that introduce these core concepts to ensure clarity and coherence.
> > - Regarding the experimental aspect, we acknowledge the value of including a broader literature review on recent applications of EBMs, particularly in physics, chemistry, and thermodynamics, given their strong conceptual ties to statistical physics. We will make an effort to incorporate a more structured overview of these applications in a Related Works section.
> >
> > We will now address the subsequent list of highlighted issues:
> >
> > - We will carefully review the manuscript trying to identify and eliminate redundancies, wordiness, and any analogies that do not significantly contribute to the clarity or depth of the discussion. The aim is to improve the readability and efficiency of the paper, ensuring that every section adds value without overburdening the reader.
> > - We will better highlight the connection with of Lemma 5.1 with Proposition 5.1, in order to be more precise in the formulation of the former.
> > - The authors acknowledge that score matching and NCE are effective methods for training EBMs, and they will add references to these techniques in the manuscript. The aim will be to integrate these methods without making the text overly verbose in comparison to the discussion on contrastive divergence.
> > - We apologize for the issues raised regarding the introduction. They understand the importance of balancing approachability with conciseness, precision, clarity, and scientific rigor. We will address each of the concerns in the detailed list provided and make the necessary revisions to improve the manuscript. Hopefully the introduction will be more effective thank to your valuable feedback.

---

### Review · Reviewer_pNBU · 2025-01-31

**Summary Of Contributions:**

The paper provides an overview of Energy-Based Models in the context of generative modeling, and shorter overviews on related generative modeling techniques: variational autoencoders, generative adversarial networks, diffusion models, and normalizing flows. Thus, the work can be used as an entry point to learn about EBMs and more generally about the development of the field of generative modeling. The work touches on several other aspects: the origin of EBMs in Hopfield networks and Boltzmann machines, an overview of the MCMC methods to sample EBMs, links to statistical physics, and a description of the contrastive divergence algorithm.

**Audience:**

Yes

**Broader Impact Concerns:**

No concerns.

**Claims And Evidence:**

Yes

**Requested Changes:**

Several (most) citations are not well formatted and should use citep instead of citet.

The meaning of Table 2 is unclear. It is hard to understand how the fields in each column are analogous, especially in the evaluation column. Unless it can be clarified, it should be removed.

The sentence “In conclusion, stochastic interpolants provide a general framework closely related to other diffusion models, such as score-based diffusion, flow matching Liu et al. (2022); Lipman et al. (2022), or Schrödinge bridge De Bortoli et al. (2021).” in page 21 needs some work and further elaboration. A sentence that starts with “In conclusion” cannot refer to concepts that have not been discussed before in the text. A couple of suggestions:

- It would be good to mention earlier on that flow matching, rectified flows and stochastic interpolants are essentially the same thing, as these other two terms are just as well-known in the literature as stochastic interpolants (if not more), and to cite Liu et al. (2022); Lipman et al. (2022) earlier on as well.
- With regards to the Schrödinger bridge algorithms, while there is a similarity, the algorithms differ: finding Schrödinger bridges is a more challenging task as it involves an optimality condition on top of matching the two marginals. It would be good to discuss that in a sentence or two.

Also at the bottom of page 21, the authors claim about stochastic interpolants that “some common issues of diffusion-based generative models persist: slow generation, dependence on hyperparameters and neural architectures, and data dependence are the primary drawbacks.” While this is true, these issues are common in all generative models, and are much more acute in most of them; after all, diffusion models and stochastic interpolants / flow matching are the  SOTA approach for most if not all settings that require data generation in continuous domains. For example, with regards to generation speed, sampling EBMs is arguably slower than sampling diffusion models as it requires using MCMC algorithms that have weak convergence guarantees. Moreover, distillation techniques for diffusion models bring down the cost of generating a sample to just a few model evaluations.

Related to my previous point, the paper does not emphasize enough the dominance of diffusion models / stochastic interpolants / flow matching as the current generative modeling techniques of choice. The authors need to comment on that, and also on the role of EBMs in the current generative modeling landscape, i.e. where should/are they be used? should/can they be combined with diffusion models? which insights can we take from EBMs?

**Strengths And Weaknesses:**

The writing style is pedagogical, in the fashion of scientific dissemination books, at risk of being too verbose in certain parts (e.g. in Section 1.1, and at the beginning of Section 3). That may be appreciated by readers that lack prior knowledge of the field or those that look for a historical perspective that is usually lacking in machine learning papers.

The paper has some weaknesses that I detail in the Requested Changes section.

---

> ### Author Response · Authors · 2025-02-03
>
> We appreciate the reviewer’s thorough evaluation of our work, as well as their insights into its strengths and areas for improvement. Below, we respond to each identified weakness, outlining possible revisions to meet the requested changes.
>
> -We will proceed to use the citep format as requested.
>
> -Table 2 was intended to summarize the key points from the previous section on generative models. However, given the paper’s already detailed discussion, removing the table may be a more effective choice than expanding it for clarity.
>
> -Regarding the content on page 21, we agree with the comment and acknowledge the overlap between flow matching, stochastic interpolation, and rectified flow. We will revise this section to better emphasize this connection, its significance for the state of the art, and its relationship with the Schrödinger Bridge, following the reviewer's suggestion on the style.
>
> -Regarding the comment from "Also at the bottom of...": we appreciate the reviewer’s insightful comments on the role of diffusion models, stochastic interpolants, and flow matching in the current generative modeling landscape. We acknowledge that the issues mentioned—such as slow generation and dependence on hyperparameters—are common to all generative models, though their severity varies. We will revise our discussion to clarify that, despite these challenges, diffusion models and their variants remain the state-of-the-art approach for continuous data generation. Regarding EBMs, we recognize the need to better contextualize their role in modern generative modeling. We will expand our discussion to highlight scenarios where EBMs can be useful, their potential synergies with diffusion-based methods, and the unique insights they offer. In particular, we will address how EBMs can contribute to aspects such as energy-based regularization, improved likelihood estimation, and alternative sampling strategies. In particular, we will highlight scenarios where obtaining a normalized probability density as the outcome of training provides an advantage over flow-based sampling, which typically does not yield explicit information about the target density. A clear example is unsupervised outlier detection or sample comparison, where access to a normalized probability density function can be crucial. Moreover, we will stress more the importance that EBMs can have in scientific ML, that is in applications related to physical systems, where the trained energy landscape can represent an additional information providing insight on the phenomenon behind the dataset at disposal.

---

### Author Response · Authors · 2025-02-04

The authors sincerely thank all the reviewers for their thorough and constructive feedback. The comments and suggestions provided are highly valuable in improving the clarity, balance, and rigor of the manuscript. A revised version of the paper, incorporating the requested changes, will be prepared as soon as possible, with all modifications highlighted in red for clarity. We appreciate your time and effort in reviewing our work.

---

### Author Response · Authors · 2025-02-14

We uploaded the modified version. We remain at disposal for interaction with the reviewers for additional modifications.

---

### Decision · Action_Editor_GeuY · 2025-04-16

**Recommendation:** Accept with minor revision

**Comment:**

The paper provides an interesting and comprehensive resource on energy-based models viewed from a physicist's point of view, along with links to other generative models. In this sense, it could be a valuable contribution to TMLR.

Nevertheless, reviewers pointed out some concerns about the manuscript in its current form, and some major changes are necessary in order to make it suitable for publication. Here are some recommendations (inspired by the reviewers' final recommendations, see below):
* There is currently a little too much broad introductory material, e.g. in Sections 1.1 and 1.2, which should be shortened. Also please include introductory text before 1.1 which clarifies the goal of the survey, ideally comparing to related surveys on EBMs, such as [this one](https://arxiv.org/abs/2101.03288). Also, move the organization bullets in p.5 to a separate section.
* Sections 3 and 4 are somewhat unrelated to the main focus on EBMs: these could be merged into a single section about links with other generative models, and significantly shortened, given that some of these (e.g., GANs and interpolants) aren't so closely related to EBMs.
* Section 7 could be expanded to cover more recent developments in EBM training, which would make the survey more valuable to the community. The term "state-of-the-art" seems somewhat misplaced to described an old method like CD, perhaps "Recent progress in EBM training" could be more appropriate?
* It would be useful to add a final section highlighting potential new directions and open problems that arise based on the proposed viewpoint, and that could benefit from more exchange between the ML and statistical physics communities.


-------
For reference, I am including here some comments by two out of three reviewers who are leaning on the negative side:
>Currently, a large part of the paper is dedicated to introducing other models, or introducing components related to EBMs (section 6, section 5), with only little content left focused on EBMs themselves, apart from section 7, which overlaps with existing review papers on training energy based models (Song et. al, 2021). Therefore, I am still leaning towards rejection. To be accepted, the paper should either:
>
> * include more new EBM-specific content (I would not hesitate to state technical results in their entirety in order to anchor clearly the properties of EBMs), and less non-EBM specific/introductory content.
> * or position itself as a general introduction to generative models (in which case, the sections related to components of EBMs should be removed)

> I think the key weakness still remains as in it's unclear what a reader can take away from a long read. As another reviewer also pointed out, there is a lot of introductory material, while the discussion on the connection between EBMs and other topics is currently NOT taking a significant enough portion of the paper. A "Hitchhiker’s guide" should articulate the relations very clearly and perhaps discuss practical implications. While introductory materials might be useful to some readers, in this reviewer's opinion, such materials may be put in an appendix and referenced when needed. The main paper can then focus on the relations/insights.

**Audience:**

Yes there is an audience who could be interested by this paper. In particular ML generative modeling people interested in learning the links with statistical physics, or physicists wanting to learn about generative models.

**Claims And Evidence:**

Yes, this is a survey that cover known topics.